# CRITIC SEQUENTIAL MONTE CARLO

**Vasileios Lioutas**[*,1,2]**, J. Wilder Lavington**[1,2]**, Justice Sefas**[1,2]**, Matthew Niedoba**[1,2]**,
Yunpeng Liu**[1,2]**, Berend Zwartsenberg**[1]**, Setareh Dabiri**[1]**, Frank Wood**[1,2,3]**, Adam Ścibior**[1]
[1]Inverted AI, [2]University of British Columbia, [3]Mila; *`vasileios.lioutas@inverted.ai`

## ABSTRACT

We introduce CriticSMC, a new algorithm for planning as inference built from a composition of sequential Monte Carlo with learned Soft-Q function heuristic factors. These heuristic factors, obtained from parametric approximations of the marginal likelihood ahead, more effectively guide SMC towards the desired target distribution, which is particularly helpful for planning in environments with hard constraints placed sparsely in time. Compared with previous work, we modify the placement of such heuristic factors, which allows us to cheaply propose and evaluate large numbers of putative action particles, greatly increasing inference and planning efficiency. CriticSMC is compatible with informative priors, whose density function need not be known, and can be used as a model-free control algorithm. Our experiments on collision avoidance in a high-dimensional simulated driving task show that CriticSMC significantly reduces collision rates at a low computational cost while maintaining realism and diversity of driving behaviors across vehicles and environment scenarios.

## 1 INTRODUCTION

Sequential Monte Carlo (SMC) (Gordon et al., 1993) is a popular, highly customizable inference algorithm that is well suited to posterior inference in state-space models (Arulampalam et al., 2002; Andrieu et al., 2004; Cappe et al., 2007). SMC is a form of importance sampling, that breaks down a high-dimensional sampling problem into a sequence of low-dimensional ones, making them tractable through repeated application of resampling. SMC in practice requires informative observations at each time step to be efficient when a finite number of particles is used. When observations are sparse, SMC loses its typical advantages and needs to be augmented with particle smoothing and backward messages to retain good performance (Kitagawa, 1994; Moral et al., 2009; Douc et al., 2011).

SMC can be applied to planning problems using the planning-as-inference framework (Ziebart et al., 2010; Neumann, 2011; Rawlik et al., 2012; Kappen et al., 2012; Levine, 2018; Abdolmaleki et al., 2018; Lavington et al., 2021). In this paper we are interested in solving planning problems with sparse, hard constraints, such as avoiding collisions while driving. In this setting, such a constraint is not violated until the collision occurs, but braking needs to occur well in advance to avoid it. Figure 1 demonstrates on a toy example how SMC requires an excessive number of particles to solve such problems. In the language of optimal control (OC) and reinforcement learning (RL), collision avoidance is a *sparse reward problem*. In this setting, parametric estimators of future rewards (Nair et al., 2018; Riedmiller et al., 2018) are learned in order to alleviate the credit assignment problem (Sutton & Barto, 2018; Dulac-Arnold et al., 2021) and facilitate efficient learning.

In this paper we propose a novel formulation of SMC, called *CriticSMC*, where a learned critic, inspired by Q-functions in RL (Sutton & Barto, 2018), is used as a *heuristic factor* (Stuhlmüller et al., 2015) in SMC to ameliorate the problem of sparse observations. We borrow from the recent advances in deep-RL (Haarnoja et al., 2018a; Hessel et al., 2018) to learn a critic which approximates future likelihoods in a parametric form. While similar ideas have been proposed in the past (Rawlik et al., 2012; Piché et al., 2019), in this paper we instead suggest (1) using soft Q-functions (Rawlik et al., 2012; Chan et al., 2021; Lavington et al., 2021) as heuristic factors, and (2) choosing the placement of such factors to allow for efficient exploration of action-space through the use of putative particles (Fearnhead, 2004). Additionally, we design CriticSMC to be compatible with informative prior distributions, which may not include an associated (known) log-density function. In planning contexts, such priors can specify additional requirements that may be difficult to define via rewards, such as maintaining human-like driving behavior.

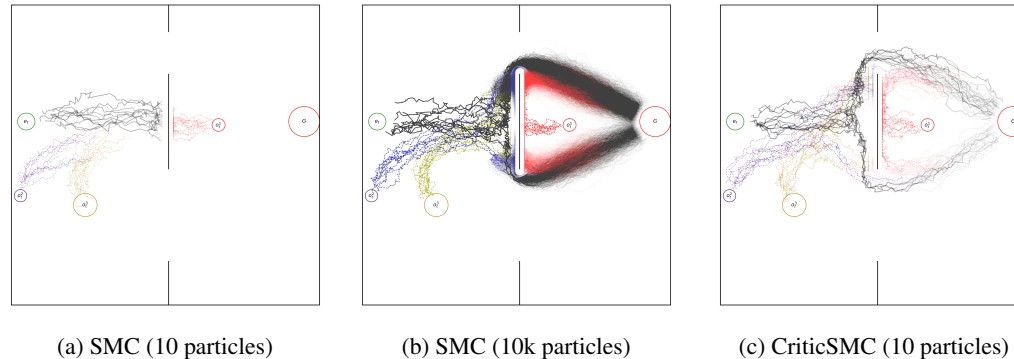

| (a) SMC (10 particles) | (b) SMC (10k particles) | (c) CriticSMC (10 particles) |

Figure 1: Illustration of the difference between CriticSMC and SMC in a toy environment in which a green ego agent is trying to reach the red goal without being hit by any of the three chasing adversaries. All plots show overlaid samples of environment trajectories conditioned on the ego agent achieving its goal. While SMC will asymptotically explore the whole space of environment trajectories, CriticSMC's method of using the critic as a heuristic within SMC encourages computationally efficient discovery of diverse high reward trajectories. SMC with a small number of particles fails here because the reward is sparse and the ego agent's prior behavioral policy assigns low probability to trajectories that avoid the barrier and the other agents.

We show experimentally that CriticSMC is able to refine the policy of a foundation (Bommasani et al., 2021) autonomous-driving behavior model to take actions that produce significantly fewer collisions while retaining key behavioral distribution characteristics of the foundation model. This is important not only for the eventual goal of learning complete autonomous driving policies (Jain et al., 2021; Hawke et al., 2021), but also immediately for constructing realistic infraction-free simulations to be employed by autonomous vehicle controllers (Suo et al., 2021; Bergamini et al., 2021; Ścibior et al., 2021; Lioutas et al., 2022) for training and testing. Planning, either in simulation or real world, requires a model of the world (Ha & Schmidhuber, 2018). While CriticSMC can act as a planner in this context, we show that it can just as easily be used for model-free online control without a world model. This is done by densely sampling putative action particles and using the critic to select amongst these sampled actions. We also provide ablation studies which demonstrate that the two key components of CriticSMC, namely the use of the soft Q-functions and putative action particles, significantly improve performance over relevant baselines with similar computational resources.

## 2 PRELIMINARIES

Since we are primarily concerned with planning problems, we work within the framework of Markov decision processes (MDPs). An MDP $\mathcal{M} = \{\mathcal{S}, \mathcal{A}, f, \mathcal{P}_0, \mathcal{R}, \Pi\}$ is defined by a set of states $s \in \mathcal{S}$, actions $a \in \mathcal{A}$, reward function $r(s, a, s') \in \mathcal{R}$, deterministic transition dynamics function $f(s, a)$, initial state distribution $p_0(s) \in \mathcal{P}_0$, and policy distribution $\pi(a|s) \in \Pi$. Trajectories are generated by first sampling from the initial state distribution $s_1 \sim p_0$, then sequentially sampling from the policy $a_t \sim \pi(a_t|s_t)$ and then the transition dynamics $s_{t+1} \leftarrow f(s_t, a_t)$ for $T$-1 time steps. Execution of this stochastic process produces a trajectory $\tau = \{(s_1, a_1), \ldots, (s_T, a_T)\} \sim p^\pi$, which is then scored using the reward function $r$. The goal in RL and OC is to produce a policy $\pi^* = \arg\max_\pi \mathbb{E}_{p^\pi}[\sum_{t=1}^T r(s_t, a_t, s_{t+1})]$. We now relate this stochastic process to inference.

### 2.1 REINFORCEMENT LEARNING AS INFERENCE

RL-as-inference (RLAI) considers the relationship between RL and approximate posterior inference to produce a class of divergence minimization algorithms able to estimate the optimal RL policy. The posterior we target is defined by a set of observed random variables $O_{1:T}$ and latent random variables $\tau_{1:T}$. Here, $O$ defines "optimality" random variables which are Bernoulli distributed with probability proportional to exponentiated reward values (Ziebart et al., 2010; Neumann, 2011; Levine, 2018). They determine whether an individual tuple $\tau_t = \{s_t, a_t, s_{t+1}\}$ is optimal ($O_t = 1$) or sub-optimal

| 1: **algorithm** STANDARD | 1: **algorithm** H-FACTORS | 1: **algorithm** CRITICSMC |
|---|---|---|
| 2: $a_t^i \sim \pi(a_t \mid s_t^i)$ | 2: $a_t^i \sim \pi(a_t \mid s_t^i)$ | 2: $a_t^i \sim \pi(a_t \mid s_t^i)$ |
| 3: $\hat{s}_{t+1}^i \leftarrow f(s_t^i, a_t^i)$ | 3: $\hat{s}_{t+1}^i \leftarrow f(s_t^i, a_t^i)$ | 3: |
| 4: $\hat{w}_t^i \leftarrow \bar{w}_{t-1}^i e^{r(s_t^i, a_t^i, \hat{s}_{t+1}^i)}$ | 4: $\hat{w}_t^i \leftarrow \bar{w}_{t-1}^i e^{r(s_t^i, a_t^i, \hat{s}_{t+1}^i) + h_t^i}$ | 4: $\hat{w}_t^i \leftarrow \bar{w}_{t-1}^i e^{h_t^i}$ |
| 5: $\alpha_t^i \sim \text{RESAMPLE}(\hat{w}_t^{1:N})$ | 5: $\alpha_t^i \sim \text{RESAMPLE}(\hat{w}_t^{1:N})$ | 5: $\alpha_t^i \sim \text{RESAMPLE}(\hat{w}_t^{1:N})$ |
| 6: $s_{t+1}^i \leftarrow \hat{s}_{t+1}^{\alpha_t^i}$ | 6: $s_{t+1}^i \leftarrow \hat{s}_{t+1}^{\alpha_t^i}$ | 6: $s_{t+1}^i \leftarrow f(s_t^{\alpha_t^i}, a_t^{\alpha_t^i})$ |
| 7: $\bar{w}_t^i \leftarrow \frac{1}{N}$ | 7: $\bar{w}_t^i \leftarrow \frac{1}{N} e^{-h_t^{\alpha_t^i}}$ | 7: $\bar{w}_t^i \leftarrow \frac{1}{N} e^{r(s_t^{\alpha_t^i}, a_t^{\alpha_t^i}, s_{t+1}^i) - h_t^{\alpha_t^i}}$ |
| 8: **end algorithm** | 8: **end algorithm** | 8: **end algorithm** |
| (a) No heuristic factors | (b) With heuristic factors | (c) CriticSMC version |

Figure 2: Main loop of SMC without heuristic factors (left), with naive heuristic factors $h_t$ (middle) and with the placement we use in CriticSMC (right). We use $\hat{w}$ for pre-resampling weights and $\bar{w}$ for post-resampling weights and we elide the normalizing factor $W_t = \sum_{i=1}^{N} \hat{w}_t^i$ for clarity. The placement of $h_t$ in CriticSMC crucially enables using putative action particles in Section 3.3.

($O_t = 0$). We replace $O_t = 1$ with $O_t$ in the remainder of the paper for conciseness. While we can rarely compute the posterior $p(s_{1:T}, a_{1:T} \mid O_{1:T})$ in closed form, we assume the joint distribution

$$p(s_{1:T}, a_{1:T}, O_{1:T}) = p_0(s_1) \prod_{t=1}^{T} p(O_t \mid s_t, a_t, s_{t+1}) \delta_{f(s_t, a_t)}(s_{t+1}) \pi(a_t \mid s_t), \tag{1}$$

where $\delta_{f(s_t, a_t)}$ is a Dirac measure centered on $f(s_t, a_t)$. This joint distribution can be used following standard procedures from variational inference to learn or estimate the posterior distribution of interest (Kingma & Welling, 2014). How close the estimated policy is to the optimal policy often depends upon the chosen reward surface, the prior distribution over actions, and chosen policy distribution class. Generally, the prior is chosen to contain minimal information in order to maximize the entropy of the resulting approximate posterior distribution (Ziebart et al., 2010; Haarnoja et al., 2018a). Contrary to classical RL, we are interested in using informative priors whose attributes we want to preserve while maximizing the expected reward ahead. In order to manage this trade-off, we now consider more general inference algorithms for state-space models.

## 2.2 SEQUENTIAL MONTE-CARLO

SMC (Gordon et al., 1993) is a popular algorithm that can be used to sample from the posterior distribution in non-linear state-space models and HMMs. In RLAI, SMC sequentially approximates the filtering distributions $p(s_t, a_t \mid O_{1:t})$ for $t \in 1 \ldots T$ using a collection of weighted samples called particles. The crucial resampling step adaptively focuses computation on the most promising particles while still producing an unbiased estimation of the marginal likelihood (Moral, 2004; Chopin et al., 2012; Pitt et al., 2012; Naesseth et al., 2014; Le, 2017). The primary sampling loop for SMC in a Markov decision process is provided in Figure 2a, and proceeds by sampling an action $a_t$ given a state $s_t$, generating the **next state $s_{t+1}$** using the environment or a model of the world, computing a weight $\hat{w}_t$ using the **reward function $r$**, and resampling from this particle population. The post-resampling weights $\bar{w}_t$ are assumed to be uniform for simplicity but non-uniform resampling schemes exist (Fearnhead & Clifford, 2003). Here, each timestep only performs simple importance sampling linking the posterior $p(s_t, a_t \mid O_{1:t})$ to $p(s_{t+1}, a_{t+1} \mid O_{1:t+1})$. When the observed likelihood information is insufficient, the particles may fail to cover the portion of the space required to approximate the next posterior timestep. For example, if all current particles have the vehicle moving at high speed towards the obstacle, it may be too late to brake and causing SMC to erroneously conclude that a collision was inevitable, while in fact it just did not explore braking actions earlier on in time.

As shown by Stuhlmüller et al. (2015), we can introduce arbitrary **heuristic factors $h_t$** into SMC before resampling, as shown in Figure 2b, mitigating the insufficient observed likelihood information. $h_t$ can be a function of anything sampled up to the point where it is introduced, does not alter the asymptotic behavior of SMC, and can dramatically improve finite sample efficiency if chosen carefully. In this setting, the optimal choice for $h_t$ is the marginal log-likelihood ahead $\sum_t^T \log p(O_{t:T} \mid s_t, a_t)$,

which is typically intractable to compute but can be approximated. In the context of avoiding collisions, this term estimates the likelihood of future collisions from a given state. A typical application of such heuristic factors in RLAI, as given by Piché et al. (2019), is shown in Figure 2b.

## 3 CRITICSMC

Historically, heuristic factors in SMC are placed alongside the reward, which is computed by taking a single step in the environment (Figure 2b). The crucial issue with this methodology is that updating weights requires computing the next state (Line 3 in Figure 2b), which can both be expensive in complex simulators, and would prevent the use in online control without a world model. In order to avoid this issue while maintaining the advantages of SMC with heuristic factors, we propose to score particles using only the heuristic factor, resample, then compute the next state and the reward, as shown in Figure 2c. We choose $h_t$ which only depends on the previous state and actions observed and not the future state, so that we can sample and score a significantly larger number of so-called putative action particles, thereby increasing the likelihood of sampling particles with a large $h_t$. In this section we first show how to construct such $h_t$, then how to learn an approximation to it, and finally how to take full advantage of this sampling procedure using putative action particles.

### 3.1 FUTURE LIKELIHOODS AS HEURISTIC FACTORS

We consider environments where planning is needed to satisfy certain hard constraints $C(s_t)$ and define the violations of such constraints as *infractions*. This makes the reward function (and thus the log-likelihood) defined in Section 2 sparse,

$$\log p(O_t|s_t, a_t, s_{t+1}) = r(s_t, a_t, s_{t+1}) = \begin{cases} 0, & \text{if } C(s_{t+1}) \text{ is satisfied} \\ -\beta_{\text{pen}}, & \text{otherwise} \end{cases} \quad (2)$$

where $\beta_{\text{pen}} > 0$ is a penalty coefficient. At every time-step, the agent receives a reward signal indicating if an infraction occurred (e.g. there was a collision). To guide SMC particles towards states that are more likely to avoid infractions in the future, we use $h_t$ which approximate *future likelihoods* (Kim et al., 2020) defined as $h_t \approx \log p(O_{t:T}|s_t, a_t)$. Such heuristic factors up-weight particles proportionally to how likely they are to avoid infractions in the future but can be difficult to accurately estimate in practice.

As has been shown in previous work (Rawlik et al., 2012; Levine, 2018; Piché et al., 2019; Lavington et al., 2021), $\log p(O_{t:T}|s_t, a_t)$ corresponds to the "soft" version of the state-action value function $Q(s_t, a_t)$ used in RL, often called the *critic*. Following Levine (2018), we use the same symbol $Q$ for the soft-critic. Under deterministic state transitions $s_{t+1} \leftarrow f(s_t, a_t)$, the soft $Q$ function satisfies the following equation, which follows from the exponential definition of the reward given in Equation 2 (a proof is provided in Section A.2 of the Appendix),

$$Q(s_t, a_t) := \log p(O_{t:T}|s_t, a_t) = r(s_t, a_t, s_{t+1}) + \log \mathbb{E}_{a_{t+1} \sim \pi(a_{t+1}|s_{t+1})} \left[ e^{Q(s_{t+1}, a_{t+1})} \right]. \quad (3)$$

CriticSMC sets the heuristic factor $h_t = Q(s_t, a_t)$, as shown in Figure 2c. We note that alternatively one could use the state value function for the next state $V(s_{t+1}) = \log \mathbb{E}_{a_{t+1}}[\exp(Q(a_{t+1}, s_{t+1}))]$, as shown in Figure 2b. This would be equivalent to the SMC algorithm of Piché et al. (2019) (see Section A.2 of the Appendix), which was originally derived using the two-filter formula (Bresler, 1986; Kitagawa, 1994) instead of heuristic factors. The primary advantage of the CriticSMC formulation is that the heuristic factor can be computed before the next state, thus allowing the application of putative action particles.

### 3.2 LEARNING CRITIC MODELS WITH SOFT Q-LEARNING

Because we do not have direct access to $Q$, we estimate it parametrically with $Q_\phi$. Equation 3 suggests the following training objective for learning the state-action critic (Lavington et al., 2021)

$$\mathcal{L}_{\text{TD}}(\phi) = \mathbb{E}_{s_t, a_t, s_{t+1} \sim d_{\text{SAO}}} \left[ \left( Q_\phi(s_t, a_t) - r(s_t, a_t, s_{t+1}) - \log \mathbb{E}_{a_{t+1} \sim \pi(a_{t+1}|s_{t+1})} \left[ e^{Q_{\perp(\phi)}(s_{t+1}, a_{t+1})} \right] \right)^2 \right]$$

$$\approx \mathbb{E}_{s_t, a_t, s_{t+1} \sim d_{\text{SAO}}} \left[ \mathbb{E}_{a_{t+1}^{1:K} \sim \pi(a_{t+1}|s_{t+1})} \left[ \left( Q_\phi(s_t, a_t) - \tilde{Q}_{\text{TA}}(s_t, a_t, s_{t+1}, \hat{a}_{t+1}^{1:K}) \right)^2 \right] \right], \quad (4)$$

where $d_{\text{SAO}}$ is the *state-action occupancy* (SAO) induced by CriticSMC, $\perp$ is the stop-gradient operator (Foerster et al., 2018) indicating that the gradient of the enclosed term is discarded, and the approximate target value $\tilde{Q}_{\text{TA}}$ is defined as

$$\tilde{Q}_{\text{TA}}(s_t, a_t, s_{t+1}, \hat{a}_{t+1}^{1:K}) = r(s_t, a_t, s_{t+1}) + \gamma \log \frac{1}{K} \sum_{j=1}^{K} e^{Q_{\perp(\phi)}(s_{t+1}, \hat{a}_{t+1}^j)}. \tag{5}$$

The discount factor $\gamma$ is introduced to reduce variance and improve the convergence of Soft-Q iteration (Bertsekas, 2019; Chan et al., 2021). For stability, we replace the bootstrap term $Q_{\perp(\phi)}$ with a $\phi$-averaging target network $Q_\psi$ (Lillicrap et al., 2016), and use prioritized experience replay (Schaul et al., 2016), a non-uniform sampling procedure. These modifications are standard in deep RL, and help improve stability and convergence of the trained critic (Hessel et al., 2018). We note that unlike Piché et al. (2019), we learn the soft-Q function for the (static) prior policy, dramatically simplifying the training process, and allowing faster sampling at inference time.

### 3.3 PUTATIVE ACTION PARTICLES

Sampling actions given states is often computationally cheap when compared to generating states following transition dynamics. Even when a large model is used to define the prior policy, it is typically structured such that the bulk of the computation is spent processing the state information and then a relatively small probabilistic head can be used to sample many actions. To take advantage of this, we temporarily increase the particle population size $K$-fold when sampling actions and then reduce it by resampling before the new state is computed. This is enabled by the placement of heuristic factors between sampling the action and computing the next state, as highlighted in Figure 2c. Specifically, at each time step $t$ for each particle $i$ we sample $K$ actions $\hat{a}_t^{i,j}$, resulting in $N \cdot K$ putative action particles (Fearnhead, 2004). The critic is then applied as a heuristic factor to each putative particle, and a population of size $N$ re-sampled following the next time-step using these weighted examples. The full algorithm is given in Algorithm 1.

---

**Algorithm 1** Critic Sequential Monte Carlo

**procedure** CRITICSMC($p_0, f, \pi, r, Q\ N, K, T$)
    Sample $s_1^{1:N} \sim p_0(s_1)$
    Set $\bar{w}_0^{1:N} \leftarrow \frac{1}{N}$
    **for** $t \in 1 \ldots T$ **do**
        **for** $n \in 1 \ldots N$ **do**
            **for** $k \in 1 \ldots K$ **do**
                Sample $\hat{a}_t^{n,k} \sim \pi(a_t|s_t^n)$
                Set $\hat{w}_t^{n \cdot N + k} \leftarrow \frac{1}{K} \bar{w}_{t-1}^n e^{Q(s_t^n, \hat{a}_t^{n,k})}$
            **end for**
        **end for**
        Set $W_t \leftarrow \sum_{i=1}^{N \cdot K} \hat{w}_t^i$
        Sample $\alpha_t^{1:N} \sim$ RESAMPLE $\left( \frac{\hat{w}_t^{1:N \cdot K}}{W_t} \right)$
        **for** $n \in 1 \ldots N$ **do**
            Set $i \leftarrow \lfloor \alpha_t^n / K \rfloor + 1$
            Set $j \leftarrow (\alpha_t^n \bmod K) + 1$
            Set $a_t^n \leftarrow \hat{a}_t^{i,j}$
            Set $s_{t+1}^n \leftarrow f(s_t^i, \hat{a}_t^{i,j})$
            Set $\bar{w}_t^n \leftarrow \frac{1}{N} W_t e^{r(s_t^i, \hat{a}_t^{i,j}, s_{t+1}^n) - Q(s_t^i, \hat{a}_t^{i,j})}$
        **end for**
    **end for**
    **return** $s_{1:T}^{1:N}, a_{1:T}^{1:N}, \bar{w}_{1:T}^{1:N}$
**end procedure**

---

For low dimensional action spaces, it is possible to sample actions densely under the prior, eliminating the need for a separate proposal distribution. This is particularly beneficial in settings where the prior policy is only defined implicitly by a sampler and its log-density cannot be quickly evaluated everywhere. In the autonomous driving context, the decision leading to certain actions can be complex, but the action space is only two- or three-dimensional. Using CriticSMC, a prior generating human-like actions can be provided as a sampler without the need for a density function. Lastly, CriticSMC can be used for model-free online control through sampling putative actions from the current state, applying the critic, and then selecting a single action through resampling. This can be regarded as a prior-aware approach to selecting actions similar to algorithms proposed by Abdolmaleki et al. (2018); Song et al. (2019).

## 4 EXPERIMENTS

We demonstrate the effectiveness of CriticSMC for probabilistic planning where multiple future possible rollouts are simulated from a given initial state using CriticSMC using two environments: a multi-agent point-mass toy environment and a high-dimensional driving simulator. In both environments infractions are defined as collisions with either other agents or the walls. Since the environment dynamics are known and deterministic, we do not learn a state transition model of the world and there

is no need to re-plan actions in subsequent time steps. We also show that CriticSMC successfully avoids collisions in the driving environment when deployed in a model-free fashion in which the proposed optimal actions are executed directly in the environment at every timestep during the CriticSMC process. Finally, we show that both the use of putative particles and the Soft-Q function instead of the standard Hard-Q result in significant improvement in terms of reducing infractions and maintaining behavior close to the prior.

## 4.1 Toy Environment

In the toy environment, depicted in Figure 1, the prior policy is a Gaussian random walk towards the goal position without any information about the position of the other agents and the barrier. All external agents are randomly placed and move adversarially and deterministically toward the ego agent. The ego agent commits an infraction if any of the following is true: 1) colliding with any of the other agents, 2) hitting a wall, 3) moving outside the perimeter of the environment. Details of this environment can be found in the Appendix.

We compare CriticSMC using 50 particles and 1024 putative action particles on the planning task against several baselines, namely the prior policy, rejection sampling with 1000 maximum trials, and the SMC method of Piché et al. (2019) with 50 particles. We randomly select 500 episodes with different initial conditions and perform 6 independent rollouts for each episode. The **prior policy** has an infraction rate of **0.84**, **rejection sampling** achieves **0.78** and **SMC of Piché et al. (2019)** yields an infraction rate of **0.14**. **CriticSMC** reduces infraction rate to **0.02**.

## 4.2 Human-like Driving Behavior Modeling

Human-like driving behavior models are increasingly used to build realistic simulation for training self-driving vehicles (Suo et al., 2021; Bergamini et al., 2021; Ścibior et al., 2021), but they tend to suffer from excessive numbers of infractions, in particular collisions. In this experiment we take an existing model of human driving behavior, ITRA (Ścibior et al., 2021), as the prior policy and attempt to avoid collisions, while maintaining the human-likeness of predictions as much as possible. The environment features non-ego agents, for which we replay actions as recorded in the INTERACTION dataset (Zhan et al., 2019). The critic receives a stack of the last two ego-centric ego-rotated birdview images (Figure 3) of size $256{\times}256{\times}3$ as partial observations of the full state. This constitutes a challenging, image-based, high-dimensional continuous control environment, in contrast to Piché et al. (2019), who apply their algorithm to low dimensional vector-state spaces in the Mujoco simulator Todorov et al. (2012); Brockman et al. (2016). The key performance metric in this experiment is the average collision rate, but we also report the average displacement error ($ADE_6$) using the minimum error across six samples for each prediction, which serves as a measure of human-likeness. Finally, the maximum final distance (MFD) metric is reported to measure the diversity of the predictions. The evaluation is performed using the validation split of the INTERACTION dataset, which neither ITRA nor the critic saw during training.

We evaluate CriticSMC on a model-based planning task against the following baselines: the prior policy (ITRA), rejection sampling with 5 maximum trials and the SMC incremental weight update rule proposed by Piché et al. (2019) using 5 particles. CriticSMC uses 5 particles and 128 putative particles, noting the computational cost of using the putative particles is negligible. We perform separate evaluation in each of four locations from the INTERACTION dataset, and for each example in the validation set we execute each method six times independently to compute the performance metrics. Table 1 shows that CriticSMC reduces the collision rate substantially more than any of the baselines and that it suffers a smaller decrease in predictive error than the SMC of Piché et al. (2019). All methods are able to maintain diversity of sampled trajectories on par with the prior policy.

Next, we test CriticSMC as a model-free control method, not allowing it to interact with the environment until an action for a given time step is selected, which is equivalent to using a single particle in CriticSMC. Specifically, at each step we sample 128 putative action particles and resample one of them based on critic evaluation as a heuristic factor. We use Soft Actor-Critic (SAC) (Haarnoja et al., 2018a) as a model-free baseline, noting that other SMC variants are not applicable in this context, since they require inspecting the next state to perform resampling. We show in Table 2 that CriticSMC is able to reduce collisions without sacrificing realism and diversity in the predictions. Here SAC does notably worse in terms of both collision rate as well as ADE. This is unsurprising as

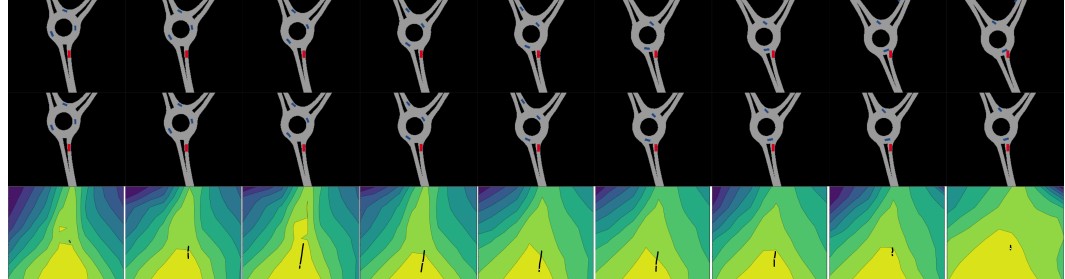

Figure 3: Collision avoidance arising from using CriticSMC for control of the red ego agent in a scenario from the INTERACTION dataset. There are three rows: the top shows the sequence of states leading to a collision arising from choosing actions from the prior policy, the middle row shows that control by CriticSMC's implicit policy avoids the collision, and the third row is a contour plot illustrating the relative values of the critic (brighter corresponds to higher expected reward) evaluated at the current state over the entire action space of acceleration (vertical axis) and steering (horizontal axis). The black dots are 128 actions sampled from the prior policy. The white dot indicates the selected action. Best viewed zoomed onscreen. For more examples see Figure 6 in the Appendix.

Table 1: Infraction rates for different inference methods performing model-predictive planning tested on four locations from the INTERACTION dataset (Zhan et al., 2019).

| Location | Method | Collision Infraction Rate | MFD | ADE$_6$ |
|---|---|---|---|---|
| DR_DEU_Merging_MT | Prior | 0.02522 | 2.2038 | 0.3024 |
| | Rejection Sampling | 0.01758 | 2.3578 | 0.3071 |
| | SMC by Piché et al. (2019) | 0.02191 | 2.3388 | 0.4817 |
| | CriticSMC | **0.01032** | 2.2009 | 0.3448 |
| DR_USA_Intersection_MA | Prior | 0.00874 | 3.1369 | 0.3969 |
| | Rejection Sampling | 0.00218 | 3.2100 | 0.3908 |
| | SMC by Piché et al. (2019) | 0.00351 | 2.8490 | 0.4622 |
| | CriticSMC | **0.00085** | 2.8713 | 0.4479 |
| DR_USA_Roundabout_FT | Prior | 0.00583 | 3.1004 | 0.4080 |
| | Rejection Sampling | 0.00133 | 3.0211 | 0.4046 |
| | SMC by Piché et al. (2019) | 0.00166 | 3.0086 | 0.4814 |
| | CriticSMC | **0.00066** | 2.9736 | 0.4439 |
| DR_DEU_Roundabout_OF | Prior | 0.00583 | 3.5536 | 0.4389 |
| | Rejection Sampling | 0.00216 | 3.4992 | 0.4287 |
| | SMC by Piché et al. (2019) | 0.00342 | 3.2836 | 0.5701 |
| | CriticSMC | **0.00083** | 3.4248 | 0.4450 |

CriticSMC takes advantage of samples from the prior, which is already performant in both metrics, while SAC must be trained from scratch. This example highlights how CriticSMC utilizes prior information more efficiently than black-box RL algorithms like SAC.

## 4.3 METHOD ABLATION

**Effect of Using Putative Action Particles** We evaluate the importance of putative action particles, via an ablation study varying the number of particles and putative particles in CriticSMC and standard SMC. Table 3 contains results that show both increasing the number of particles and putative articles have a significant impact on performance. Putative particles are particularly important since a large number of them can typically be generated with a small computational overhead.

**Comparison of Training the Critic With the Soft-Q and Hard-Q Objective** We compare the fitted Q iteration (Watkins & Dayan, 1992), which uses the maximum over Q at the next stage to update the critic (i.e., $\max_{a_{t+1}} Q(s_{t+1}, a_{t+1})$), with the fitted soft-Q iteration used by CriticSMC (Eq. 4). The results, displayed in Table 4, show that the Hard-Q heuristic factor leads to a significant reduction in collision rate over the prior, but produces a significantly higher ADE$_6$ score. We attribute this to the risk-avoiding behavior induced by hard-Q.

Table 2: Infraction rates for performing model-free online control against the prior and SAC policies tested on four locations from the INTERACTION dataset (Zhan et al., 2019).

| Location | Method | Collision Infraction Rate | MFD | ADE$_6$ |
|---|---|---|---|---|
| DR_DEU_Merging_MT | Prior | 0.02522 | 2.2038 | 0.3024 |
| | SAC | 0.03899 | 0.0 | 1.1548 |
| | CriticSMC | **0.01376** | 2.1985 | 0.3665 |
| DR_USA_Intersection_MA | Prior | 0.00874 | 3.1369 | 0.3969 |
| | SAC | 0.02700 | 0.0 | 4.1141 |
| | CriticSMC | **0.00285** | 2.9595 | 0.4641 |
| DR_USA_Roundabout_FT | Prior | 0.00583 | 3.1004 | 0.4080 |
| | SAC | 0.04501 | 0.0 | 1.7987 |
| | CriticSMC | **0.00183** | 3.0125 | 0.4567 |
| DR_DEU_Roundabout_OF | Prior | 0.00583 | 3.5536 | 0.4389 |
| | SAC | 0.06400 | 0.0 | 3.4583 |
| | CriticSMC | **0.00233** | 3.5173 | 0.4459 |

Table 3: Infraction rates for SMC and CriticSMC with a varying number of particles and putative particles, tested on 500 random initial states using the proposed toy environment.

| Method | Putative Particles | Particles | | | | |
|---|---|---|---|---|---|---|
| | | 1 | 5 | 10 | 20 | 50 |
| SMC | 1 | 0.774 | 0.488 | 0.383 | 0.288 | 0.183 |
| CriticSMC | 1 | 0.774 | 0.368 | 0.298 | 0.162 | 0.072 |
| SMC | 1024 | 0.772 | 0.415 | 0.281 | 0.179 | 0.119 |
| CriticSMC | 1024 | 0.094 | 0.031 | 0.021 | 0.016 | **0.008** |

**Execution Time Comparison** Figure 4 shows the average execution time it takes to predict 3 seconds into the future given 1 second of historical observations for the driving behavior modeling experiment. This shows that the run-time of all algorithms is of the same order, while the collision rate of CriticSMC is significantly lower, demonstrating the low overhead of using putative action particles.

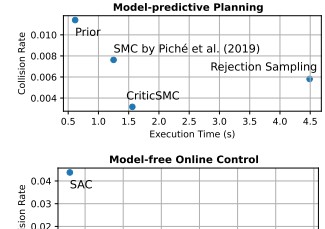

Figure 4: Execution time comparison between the baseline methods and CriticSMC for both model-predictive planning and model-free online control. The collision infraction rate is averaged across the 4 INTERACTION locations.

## 5 RELATED WORK

SMC methods (Gordon et al., 1993; Kitagawa, 1996; Liu & Chen, 1998), also known as particle filters, are a well-established family of inference methods for generating samples from posterior distributions. Their basic formulations perform well on the filtering task, but poorly on smoothing (Godsill et al., 2004) due to particle degeneracy. These issues are usually addressed using backward simulation (Lindsten & Schön, 2013) or rejuvenation moves (Gilks & Berzuini, 2001; Andrieu et al., 2010). These solutions improve sample diversity but are not sufficient in our context, where normal SMC often fails to find even a single infraction-free sample. Lazaric et al. (2007) used SMC for learning actor-critic agents with continuous action environments. Similarly to CriticSMC, Piché et al. (2019) propose using the value function as a backward message in SMC for planning. Their method is equivalent to what is obtained using the equations from Figure 2b with $h_t = V(s_{t+1}) = \log \mathbb{E}_{a_{t+1}}[\exp(Q(a_{t+1}, s_{t+1}))]$ (see proof in Section A.2 of the Appendix). This formulation cannot accommodate putative action particles and learns a parametric policy alongside $V(s_{t+1})$, instead of applying the soft Bellman update (Asadi & Littman, 2017; Chan et al., 2021) to a fixed prior.

In our experiments we used the bootstrap proposal (Gordon et al., 1993), which samples from the prior model, but in cases where the prior density can be efficiently computed, using a better proposal distribution can bring significant improvements. Such proposals can be obtained in a variety of

Table 4: Infraction rates for the Hard-Q and the Soft-Q objectives tested on the location DR_DEU_Merging_MT from the INTERACTION dataset (Zhan et al., 2019).

| Method | Critic Objective | Collision Infraction Rate | MFD | ADE$_6$ | Progress |
|--------|-----------------|---------------------------|-----|---------|----------|
| Prior | - | 0.02522 | 2.2038 | 0.3024 | 16.43 |
| CriticSMC | Hard-Q | 0.01911 | 0.9383 | 1.0385 | 14.98 |
| | Soft-Q | **0.01376** | 2.1985 | 0.3665 | 15.91 |

ways, including using unscented Kalman filters (van der Merwe et al., 2000) or neural networks minimizing the forward Kullback-Leibler divergence (Gu et al., 2015). CriticSMC can accommodate proposal distributions, but even when the exact smoothing distribution is used as a proposal, backward messages are still needed to avoid the particle populations that focus on the filtering distribution.

As we show in this work, CriticSMC can be used for planning as well as model-free online control. The policy it defines in the latter case is not parameterized explicitly, but rather obtained by combining the prior and the critic. This is similar to classical Q-learning (Watkins & Dayan, 1992), which obtains the implicit policy by taking the maximum over all actions of the Q function in discrete action spaces. This approach has been extended to continuous domains using ensembles (Deisenroth & Rasmussen, 2011; Ryu et al., 2020; Lee et al., 2021) and quantile networks (Bellemare et al., 2017). The model-free version of CriticSMC is also very similar to soft Q-learning described as described by Haarnoja et al. (2017); Abdolmaleki et al. (2018), and analyzed by Chan et al. (2021).

Imitating human driving behavior has been successful in learning control policies for autonomous vehicles (Bojarski et al., 2016; Hawke et al., 2019) and to generate realistic simulations (Bergamini et al., 2021; Ścibior et al., 2021). In both cases, minimizing collisions, continues to present one of the most important issues in autonomous vehicle research. Following a data-driven approach, Suo et al. (2021) proposed auxiliary losses for collision avoidance, while Igl et al. (2022) used adversarially trained discriminators to prune predictions that are likely to result in infractions. To the best of our knowledge, ours is the first work to apply a critic targeting the backward message in this context.

## 6 DISCUSSION

CriticSMC increases the efficiency of SMC for planning in scenarios with hard constraints, when the actions sampled must be adjusted long before the infraction takes place. It achieves this efficiency through the use of a learned critic which approximates the future likelihood using putative particles that densely sample the action space. The performance of CriticSMC relies heavily on the quality of the critic and in this work we display how to take advantage of recent advances in deep RL to obtain one. One avenue for future work is devising more efficient algorithms for learning the soft Q function such as proximal updates (Schulman et al., 2017) or the inclusion of regularization which guards against deterioration of performance late in training (Kumar et al., 2020).

The design of CriticSMC is motivated by the desire to accommodate implicit priors defined as samplers, such as the ITRA model (Ścibior et al., 2021) we used in our self-driving experiments. For this reason, we avoided learning explicit policies to use as proposal distributions since maintaining similarity with the prior can be extremely complicated. Where the prior density can be computed, learned policies could be successfully accommodated. This is particularly important when the action space is high-dimensional and it is difficult to sample it densely using putative particles.

In this work, we focused on environments with deterministic transition dynamics but CriticSMC could also be applied when dynamics are stochastic (i.e. $s_{t+1} \sim p(s_{t+1}|s_t, a_t)$). In these settings, the planning as inference framework suffers from optimism bias (Levine, 2018; Chan et al., 2021), even when exact posterior can be computed, which is usually mitigated by carefully constructing the variational family. For applications in real-world planning, CriticSMC relies on having a model of transition dynamics and the fidelity of that model is crucial for achieving good performance. Learning such models from observations is an active area of research (Ha & Schmidhuber, 2018; Chua et al., 2018; Nagabandi et al., 2020). Finally, we focused on avoiding infractions, but CriticSMC is applicable to planning with any reward surfaces and to sequential inference problems more generally.

ACKNOWLEDGMENTS

We acknowledge the support of the Natural Sciences and Engineering Research Council of Canada (NSERC), the Canada CIFAR AI Chairs Program, and the Intel Parallel Computing Centers program. Additional support was provided by UBC's Composites Research Network (CRN), and Data Science Institute (DSI). This research was enabled in part by technical support and computational resources provided by WestGrid (www.westgrid.ca), Compute Canada (www.computecanada.ca), and Advanced Research Computing at the University of British Columbia (arc.ubc.ca).

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

# A APPENDIX

## A.1 SEQUENTIAL MONTE CARLO

Here we briefly give an overview of the sequential Monte Carlo (SMC) algorithm adapted for Markov decision processes (MDPs). We borrow the notation from Section 2 in the main paper. Obtaining state-action pairs $(s_{1:T}, a_{1:t})$ that maximize the expected sum of rewards corresponds to sampling state-action pairs from the posterior $p(s_{1:T}, a_{1:T}|O_{1:T})$. The SMC inference algorithm (Gordon et al., 1993) approximate the filtering distributions $p(s_t, a_t|O_{1:t})$. In general, SMC is assuming the existence of a proposal distribution $q(s_{t+1}, a_{t+1}|s_t, a_t)$ but for simplicity we instead use bootstrap proposals that use the prior policy. The algorithm samples $N$ independent particles from the initial distribution $s_1^n \sim p_0(s_1)$ where each particle has uniform weights $w_0^n = 1/N$. At each iteration, the algorithm advances each particle one step forward by sampling actions from $\hat{a}_t^n \sim \pi(a_t|s_t^n)$ and then compute the next states by $\hat{s}_{t+1}^n \sim p(s_{t+1}|s_t^n, \hat{a}_t^n)$ accumulating the optimality likelihoods $p(O_t^n|s_t^n, \hat{a}_t^n, \hat{s}_{t+1}^n) = e^{r(s_t^n, \hat{a}_t^n, \hat{s}_{t+1}^n)}$ in the corresponding particle weight, saving the sum of weights and normalizing them before proceeding to the next time step. In our setting, we assume the state dynamics $p(s_{t+1}|s_t, a_t)$ of the environment to be deterministic $s_{t+1} \leftarrow f(s_t, a_t)$. SMC suffers from weight disparity which can lead to a reduced effective sample size of particles. This is mitigated by introducing a resampling step $\text{RESAMPLE}(\bar{w}_t^{1:N})$ at every iteration to help SMC select promising particles with high weights that have higher chance of surviving whereas particles with low weights most likely will get discarded. See Douc & Cappe (2005) for an extensive overview of different resampling schemes. Algorithm 2 summarizes the SMC process for MPDs using bootstrap proposals.

---

**Algorithm 2** Sequential Monte Carlo

---

**procedure** SMC($p_0, f, \pi, r\ N, T$)
    Sample $s_1^{1:N} \sim p_0(s_1)$
    Set $\bar{w}_0^{1:N} \leftarrow \frac{1}{N}$
    **for** $t \in 1 \dots T$ **do**
        **for** $n \in 1 \dots N$ **do**
            Sample $\hat{a}_t^n \sim \pi(a_t|s_t^n)$
            Set $\hat{s}_{t+1}^n \leftarrow f(s_t^n, \hat{a}_t^n)$
            Set $\hat{w}_t^n \leftarrow \bar{w}_{t-1}^n e^{r(s_t^n, \hat{a}_t^n, \hat{s}_{t+1}^n)}$
        **end for**
        Set $W_t \leftarrow \sum_{i=1}^N \hat{w}_t^i$
        Sample $\alpha_t^{1:N} \sim \text{RESAMPLE}\left(\frac{\hat{w}_t^{1:N}}{W_t}\right)$
        **for** $n \in 1 \dots N$ **do**
            Set $a_t^n \leftarrow \hat{a}_t^{\alpha_t^n}$
            Set $s_{t+1}^n \leftarrow \hat{s}_{t+1}^{\alpha_t^n}$
            Set $\bar{w}_t^n \leftarrow \frac{1}{N} W_t$
        **end for**
    **end for**
    **return** $s_{1:T}^{1:N}, a_{1:T}^{1:N}, \bar{w}_{1:T}^{1:N}$
**end procedure**

---

## A.2 DERIVATIONS

**Soft-Q function** Below is the derivation of the soft-Q function defined as the log probability of the backward message.

$$\begin{aligned} Q(s_t, a_t) &:= \log p(O_{t:T}|s_t, a_t) \\ &= \log p(O_t|s_t, a_t) + \log p(O_{t+1:T}|s_t, a_t), \end{aligned} \tag{6}$$

where

$$\log p(O_t|s_t, a_t) = \underset{s_{t+1} \sim p(s_{t+1}|s_t, a_t)}{\mathbb{E}} \left[ r(s_t, a_t, s_{t+1}) \right], \tag{7}$$

and

$$\log p(O_{t+1:T}|s_t, a_t) = \log \int_{s_{t+1}} \int_{a_{t+1}} p(s_{t+1}|s_t, a_t)\pi(a_{t+1}|s_{t+1})p(O_{t+1:T}|s_{t+1}, a_{t+1})da_{t+1}ds_{t+1}$$

$$= \log \mathbb{E}_{s_{t+1} \sim p(s_{t+1}|s_t, a_t)} \left[ \mathbb{E}_{a_{t+1} \sim \pi(a_{t+1}|s_{t+1})} \left[ e^{Q(s_{t+1}, a_{t+1})} \right] \right]. \tag{8}$$

If we assume the dynamics $p(s_{t+1}|s_t, a_t)$ of the environment to be deterministic $s_{t+1} \leftarrow f(s_t, a_t)$, we can simplify Equation 6 to

$$Q(s_t, a_t) = r(s_t, a_t, s_{t+1}) + \log \mathbb{E}_{a_{t+1} \sim \pi(a_{t+1}|s_{t+1})} \left[ e^{Q(s_{t+1}, a_{t+1})} \right]. \tag{9}$$

**SMC using value function as heuristic factors**   Piché et al. (2019) proposed using state values $V(s_t)$ as backward messages in SMC for planning. Based on the two-filter formula (Bresler, 1986; Kitagawa, 1994), they derive the following weight update rule

$$w_t = w_{t-1} \mathbb{E}_{s_{t+1} \sim p(s_{t+1}|s_t, a_t)} \left[ \exp \left( r(s_t, a_t, s_{t+1}) + V(s_{t+1}) - \log \mathbb{E}_{s_t \sim p(s_t|s_{t-1}, a_{t-1})} \left[ \exp \left( V(s_t) \right) \right] \right) \right]. \tag{10}$$

We omit the term $-\log \pi_\theta(a_t|s_t)$ since we assume to use bootstrap proposals instead of learning them. Thus, the current action is sampled as $a_t \sim \pi(a_t|s_t)$. In our framework, for simplicity, we assume the use of deterministic state transition dynamics $p(s_{t+1}|s_t, a_t)$ which simplifies the update rule to

$$w_t = w_{t-1} \exp \left( r(s_t, a_t, s_{t+1}) + V(s_{t+1}) - V(s_t) \right). \tag{11}$$

Piché et al. (2019) in practice trained an SAC-based policy and used the learned state-action value functions $Q(s_t, a_t)$ to approximate state values $V(s_t)$. Following a similar experimentation setting, we use a soft approximation of the value function terms using state-action value functions $Q(s_t, a_t)$ as described by Levine (2018) using

$$V(s_t) = \log \mathbb{E}_{a_t \sim \pi(a_t|s_t)} \left[ \exp(Q(s_t, a_t)) \right]. \tag{12}$$

This results in the following particle weight update rule

$$w_t = w_{t-1} \exp \left( r(s_t, a_t, s_{t+1}) + \log \mathbb{E}_{\hat{a}_{t+1} \sim \pi(a_{t+1}|s_{t+1})} \left[ \exp(Q(s_{t+1}, \hat{a}_{t+1})) \right] - \log \mathbb{E}_{\hat{a}_t \sim \pi(a_t|s_t)} \left[ \exp(Q(s_t, \hat{a}_t)) \right] \right). \tag{13}$$

We can then define the heuristic factor used in Figure 2b of the main paper as

$$h_t = \log \mathbb{E}_{\hat{a}_{t+1} \sim \pi(a_{t+1}|s_{t+1})} \left[ \exp(Q(s_{t+1}, \hat{a}_{t+1})) \right] - \log \mathbb{E}_{\hat{a}_t \sim \pi(a_t|s_t)} \left[ \exp(Q(s_t, \hat{a}_t)) \right] \tag{14}$$

which utilizes a soft approximation of the value function. It is worth emphasising that a next state sample $s_{t+1}$ from the environment model is required which makes the use of putative action particles (see Section 3.3 of the main paper) inefficient and expensive contrary to the proposed CriticSMC method.

A.3   TOY ENVIRONMENT EXPERIMENT DETAILS

In this environment, the ego agent is described by $e_t = (x_t^e, y_t^e, r^e)$ where $x, y$ is the position in the square coordinate system $[0, 1]^2$ and $r^e$ is the radius. We randomly position other agents $o_t^i = (x_t^{o^i}, y_t^{o^i}, r^{o^i})$ where $i \in [0, 5]$. In addition, there is a partial barrier in the middle with gates $g^k = (x^{g^k}, y^{g^k}, w^{g^k})$ where $x^{g^k}, y^{g^k}$ are the coordinates of the center of the gate $k$, $w^{g^k}$ is the width of the opening and $k \in [1, 3]$. Finally, a goal position $G = (x^G, y^G, r^G)$ is positioned on the other side of the barrier. The ego and the other agents are moving by displacement actions $a_t = (\Delta x_t, \Delta y_t)$.

The state representation consists of relative distances between the ego agent and the other agents, the center of the gates and the goal position. A two-layer fully connected neural network with a ReLU

activation function and size of 64 takes as input this representation and produces a state encoding. A similar network takes as input the two-dimensional displacement actions and produces the action encoding. Finally, another two-layer network takes as input the concatenation of the state and actions encodings and produce the $Q$ values.

We train the model using a single Nvidia RTX 2080Ti GPU. The prioritized experience replay buffer has a size of 1 million stored experiences. The discount factor is set to 0.99, the batch size to 256 and the learning rate to 0.001. Finally, we sample 1024 actions during running CriticSMC while training the critic model.

## A.4 Driving Behavior Model Experiment Details

The prior model we picked for this experiment is ITRA (Ścibior et al., 2021) but any other probabilistic behavior model can be used. We follow the same architecture and training procedure as described in Ścibior et al. (2021). The prior model is trained on the INTERACTION (Zhan et al., 2019) dataset and the task is that given 10 timesteps of observed behavior, predict the next 30 timesteps of future trajectories. For the critic, we used the same convolution neural network architecture as the prior model. The critic takes as input the last two observed birdviews images and encodes them separately. The concatenation of the two representations along with the action encoding is processed by a final layer that produces the $Q$ value. The architecture for these layers is the same as in Section A.3.

We train the critic model using a single Nvidia RTX 2080Ti GPU. The prioritized experience replay buffer has a size of 1.5 million stored experiences. The discount factor is set to 0.99, the batch size to 256 and the learning rate to 0.001. Finally, we sample 128 actions during running CriticSMC while training the critic model.

### A.4.1 Reinforcement Learning Environment

The environment used to train the RL agents takes as input a location from the INTERACTION dataset and trains a single-agent policy where all non-ego actors rollout according to ground truth. Because the CriticSMC algorithm rolls out every agent according to ground truth for the first ten frames of each trajectory before prediction, we simply remove these frames and begin executing the policy on frame eleven. At time step $t$, the policy takes the previous and current birdview images $(b_{t-1}, b_t)$ where each image has a size $256 \times 256 \times 1$. The stacked birdview images make the total input for the policy and value function to be $2 \times 256 \times 256 \times 1$. The policy produces an action $a_t \in [-1, 1]^2$ which corresponds to the bicycle kinematic model's relative action space (see Ścibior et al. (2021) for more details). The differentiable simulator (Ścibior et al., 2021) then uses $a_t$ to update its state and returns the next birdview image $b_{t+1}$. In this setting the policy distribution that is learned follows a squashed normal distribution (Haarnoja et al., 2018b), as is standard for the SAC implementations (Haarnoja et al., 2018b). The stochastic policy learned by SAC is tailored towards exploration and thus behaves poorly. For this reason we only report its deterministic behavior (e.g. the mode of the policy) in Table 2 of the main paper. For each of the four locations that were evaluated, the RL agents were run over three different learning rate schedules and three different reward structures for a minimum of 150k time steps. The policy uses the same convolutional neural network architecture as in CriticSMC and is updated according to the soft actor-critic algorithm in stable-baselines3 (Brockman et al., 2016). Table 5 shows the hyper-parameter settings used for training.

**Reward Surfaces** In the three rewards settings which we tested, there were a number of different feedback mechanisms which were used to produce the desired behavior (i.e. low-collision probability and low ADE). The first, was a score based reward upon an estimate of the log-probability under ITRA. To compute this "score reward", the environment passes the pair of birdview images $(b_t, b_{t+1})$ to ITRA, which generates the hypothetical action $a_t^{\text{ITRA}}$ that ITRA would have taken to make the state transition from $b_t$ to $b_{t+1}$. Then, the environment sets the reward to be a monotonic function of the likelihood of $a_t$ under a normal distribution centred around $a_t^{\text{ITRA}}$: $r_{t+1} \equiv \tanh\left(\log p(a_t; a_t^{\text{ITRA}}, \Sigma)\right)$ where $p(\cdot; a_t^{\text{ITRA}}, \Sigma) = \mathcal{N}(a_t^{\text{ITRA}}, \Sigma)$ for some covariance $\Sigma$.

Next, we include five simpler reward surfaces which have been shown to improve performance in the literature (Reda et al., 2020). First, the "action reward" is a linear function of the absolute difference between the action output by the policy and the action which ITRA would have taken at time step $t$:

| Reinforcement Learning Baseline Parameters | | |
|---|---|---|
| Parameter Name | Parameter Value(s) | Parameter Description |
| $\Sigma$ | $I_2$ | The covariance matrix of the multivariate normal distribution centred around hypothetical ITRA action $a_t^{\text{ITRA}}$. |
| $\alpha_1$ | 0.15 | Coefficient for score reward, this parameter scales how closely the agent should track estimated log-likelihood of actions under the ITRA model. |
| $\alpha_2$ | 2 | Coefficient for action reward, this incentives the policy to be as close to the mode of ITRA as possible without access to a score function over those actions. |
| $\alpha_3$ | 0.05 | Coefficient for action difference reward, this incentivizes the agent to produce sequences of actions which are smoother, and therefore often more human-like. |
| $\alpha_{4,5,6}$ | 0 or 1 | Boolean coefficients selecting whether infraction, survival, or ground truth rewards are used. |
| $\gamma$ | 0.99 | Discount factor, set to encourage lower variance gradient estimates, but greedier policy behavior (Sutton & Barto, 2018). |
| Learning Rate | 0.0002, 0.00012, 0.00008 | Learning rate for optimization (in this case the Adam Optimizer). |
| Batch Size | 256 | Number of examples used in each gradient decent update for both the critic and policy networks. |
| Buffer Size | 500000 | Size of SAC experience buffer (equivalent to maximum number of steps which can be taken within the environment). |
| Learning Starts | 1000 | Number of exploration steps used (e.g. a uniform distribution over actions) before learned stochastic policy is used to gather interactions. |
| $\tau$ | 0.005 | Polyak parameter averaging coefficient which improves convergence of deep Q learning algorithms (Haarnoja et al., 2018b). |
| Latent-Features | 256 | Number of neurons used in the output of the feature encoder, and which is fed to the standard two layer multi-layer perceptron defined by standard SAC algorithms (Haarnoja et al., 2018b). |

Table 5: Hyper-parameters for the reinforcement learning baseline used in Section 4.2. All hyper-parameters which were not listed above, use the default values provided by the SAC implementation of stable-baselines3 (Brockman et al., 2016).

$\left(2 - ||a_t - a_t^{\text{ITRA}}||_1\right)$. Second, the action difference reward is the scaled absolute difference between the current and previous actions: $||a_t - a_{t-1}||_1$. Third, the environment computes the "ground-truth reward" $r_{t+1}$ by evaluating $s_t$ against the ground truth data from the INTERACTION dataset. In particular, the environment sets the reward to be a linear function of the negative Euclidean distance at time $t+1$ between the xy-coordinate of the ego-vehicle according to the simulator, $s_{t+1}$, and that according to ground truth, $s_{t+1}^{\text{GT}}$: $100 - ||s_{t+1} - s_{t+1}^{\text{GT}}||_2$. Fourth, we include a "survival reward" of 1 if the agent does not commit an infraction at step $t$. Lastly, the infraction reward is -5 if the agent commits any type of infraction at step $t$ and 5 otherwise.

Using these five feedback mechanisms, we consider three different reward surfaces. Each of which are defined following reward calculation:

$$r_{t+1} = \alpha_1 r_{t+1}^{\text{SCORE}} + \alpha_2 r_{t+1}^{\text{ACTION}} + \alpha_3 r_{t+1}^{\text{ACTION DIFF}}$$
$$+ \alpha_4 r_{t+1}^{\text{INFRACTION}} + \alpha_5 r_{t+1}^{\text{SURVIVE}} + \alpha_6 r_{t+1}^{\text{GROUND TRUTH}} \tag{15}$$

In the first reward setting which was considered, we set all coefficients $\alpha_i$ to zero except the SURVIVE reward, and thus refer to this reward type as the survival reward setting. Next we consider a setting where we set all $\alpha_i$ to zero except the GROUND TRUTH reward, and refer to this setting as the ground-truth setting. Lastly, we considered a setting where: $\alpha_1 = 0.15$, $\alpha_2 = 2.0$, $\alpha_3 = 0.05$, and the remaining $\alpha_i$ are all set to zero. We refer to this setting as the ITRA setting, as it includes the most information about the ITRA model. To arrive at the final result, models where trained under all three of these settings, evaluated, and then chosen based upon the lowest collision infraction rate.

## A.5 CRITICSMC AS AN EFFICIENT SMC INFERENCE ALGORITHM

We include in the supplementary material a demo code implementation of CriticSMC applied to the following linear Gaussian state-space model (LGSSM) with well-defined critic function

$$f(s_t, a_t) := s_t + a_t \tag{16}$$
$$p(s_0) = \mathcal{N}(0, 1) \tag{17}$$
$$p(a_t|s_t) = \mathcal{N}(0.5 * s_t, 1) \tag{18}$$
$$p(s_{t+1}|s_t, a_t) = \delta_{f(s_t, a_t)}(s_{t+1}) \tag{19}$$
$$\log p(O_t|s_t, a_t, s_{t+1}) = \begin{cases} 0, & \text{if } -1 \times 10^{-2} \leq s_{t+1} \leq 1 \times 10^{-2} \\ -10000, & \text{otherwise} \end{cases} \tag{20}$$
$$= Q(s_t, a_t) \tag{21}$$
$$\approx -1000|s_t + a_t| + \epsilon \tag{22}$$

where the state transition function $f(s_t, a_t)$ is assumed to be computationally expensive. The conditional posterior samples from $p(s_{1:T}, a_{1:T}|O_{1:T})$ are defined as states that are within the range defined in Equation 20. We use $T = 10$ in our experiments.

Figure 5 demonstrates the performance of CriticSMC compared to SMC for estimating the (negative) log-marginal likelihood $p(O_{1:T})$ relative to the computational time needed to execute the inference algorithm.

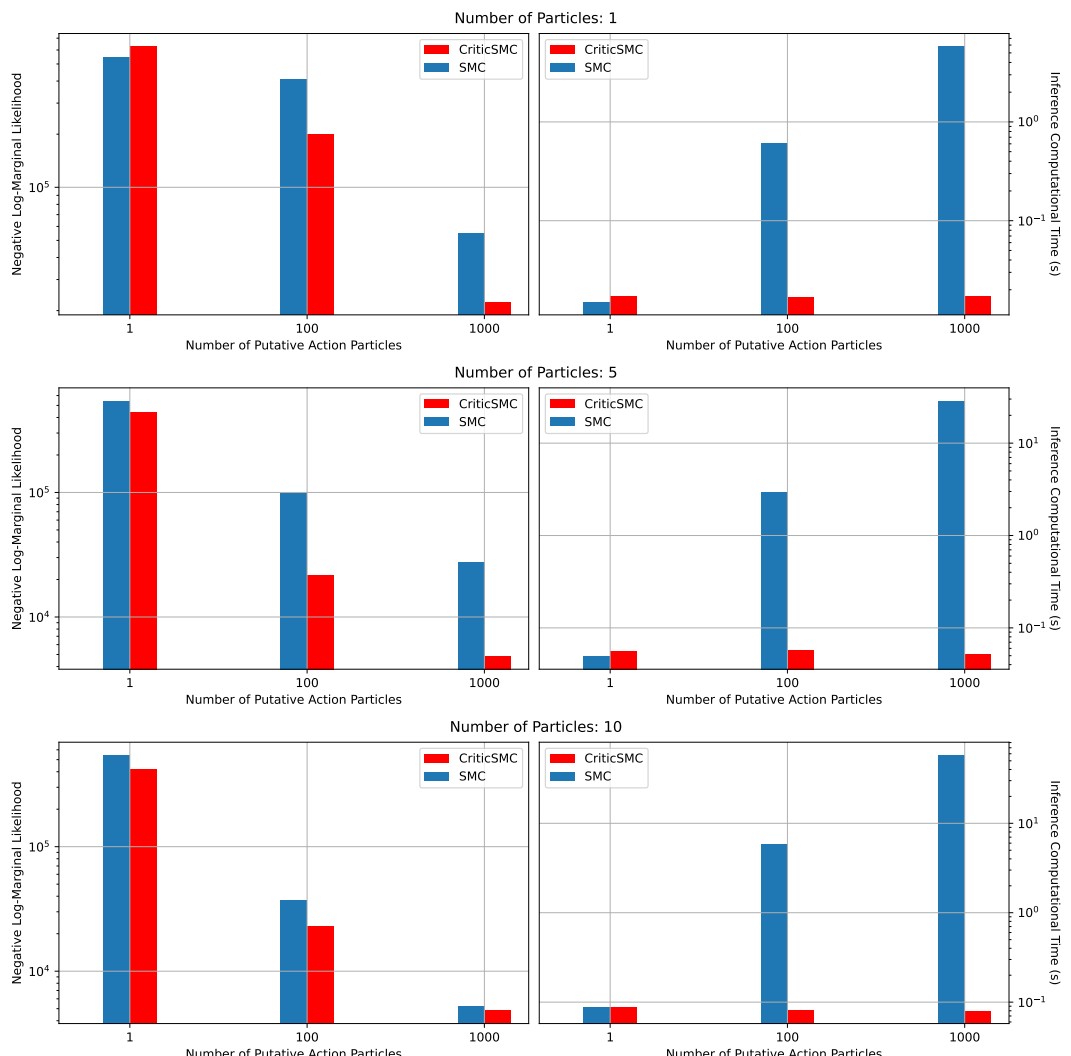

Figure 5: Given a well-defined linear Gaussian state-space model, we evaluate the performance of SMC and CriticSMC estimating the (negative) log-marginal likelihood (bar plots on the left column) relative to the speed of inference measured in wall-clock time (bar plots on the right column). We pick the number of particles as $N \in \{1, 5, 10\}$ and the number of putative action particles as $K \in \{1, 100, 1000\}$. CriticSMC is able to better estimate the marginal likelihood significantly faster than SMC by taking advantage of a large population of putative action particles and a computationally efficient critic function used as a heuristic factor to guide inference.

## A.6 NOTATIONS AND ABBREVIATIONS

| | | |
|---:|:---:|:---|
| $s_{1:T}$ | $\triangleq$ | sequence of states |
| $a_{1:T}$ | $\triangleq$ | sequence of actions |
| $\pi(a_t|s_t)$ | $\triangleq$ | prior policy |
| $p(s_{t+1}|s_t, a_t)$ | $\triangleq$ | state transition dynamics density |
| $r(s_t, a_t, s_{t+1})$ | $\triangleq$ | reward value received at timestep $t$ |
| $p(O_t|s_t, a_t, s_{t+1})$ | $\triangleq$ | optimality probability defined as the exponentiated reward |
| $Q(s_t, a_t)$ | $\triangleq$ | soft state-action value function referred to as the critic |
| $Q_\phi$ | $\triangleq$ | parametric approximation of the critic |
| $Q_\psi$ | $\triangleq$ | fixed target critic model used for computing the TD error |
| $h_t$ | $\triangleq$ | heuristic factor at timestep $t$ |
| $\hat{w}_t$ | $\triangleq$ | pre-resampling particle weight |
| $\bar{w}_t$ | $\triangleq$ | post-resampling particle weight |
| $W_t$ | $\triangleq$ | normalizing factor |
| $\alpha_t^i$ | $\triangleq$ | ancestral indices for each particle $i$ |
| $\beta_{\mathrm{pen}}$ | $\triangleq$ | penalty coefficient |
| $\gamma$ | $\triangleq$ | discount factor |
| $T$ | $\triangleq$ | horizon length |
| $t \in 1 \ldots T$ | $\triangleq$ | timesteps |
| $n \in 1 \ldots N$ | $\triangleq$ | particle number |
| $k \in 1 \ldots K$ | $\triangleq$ | putative action particle number |

Table 6: Notations

| | |
|---:|:---:|
| SMC: | Sequential Monte Carlo |
| MDP: | Markov Decision Process |
| RL: | Reinforcement Learning |
| HMM: | Hidden Markov Model |
| RLAI: | Reinforcement Learning as Inference |
| TD: | Temporal Difference |
| SAC: | Soft Actor Critic |
| MPC: | Model Predictive Control |
| ADE: | Average Displacement Error |
| MFD: | Maximum Final Distance |

Table 7: Abbreviations

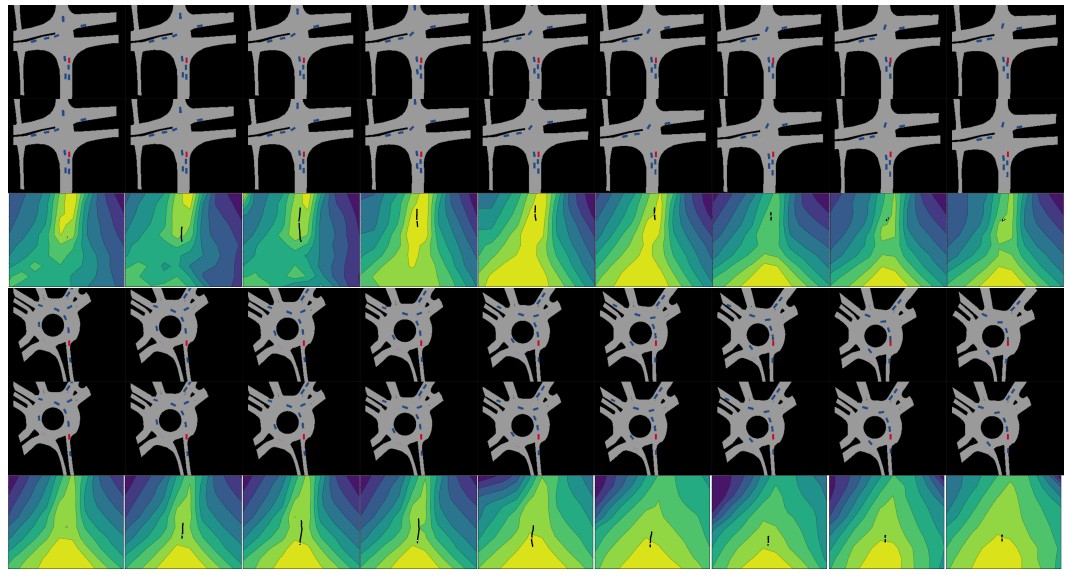

Figure 6: More examples similar to Figure 3 of the main paper.

