# OpenReview forum: "Critic Sequential Monte Carlo"
_ICLR.cc/2023/Conference — ICLR 2023 poster_

### Official Review · Reviewer_9uTg · 2022-10-14

**Confidence:** 4
**Correctness:** 4
**Technical Novelty And Significance:** 3
**Empirical Novelty And Significance:** 3
**Recommendation:** 6

**Clarity, Quality, Novelty And Reproducibility:**

The algorithm described in the paper (see: Algorithm 1) suffers from the well-known optimism bias of the control-as-inference framework [2]. This is when conditioning on optimality causes the planner to assume that any stochasticity in the system to turn out in the agent's favor. The authors are aware of this problem, and yet the only discussion of this happens in the final paragraph of the paper. Here, the authors point out that this issue does not affect the experiments in the paper because the systems used are deterministic. I have two issues here:

* The way the optimism bias is discussed is not ideal, because a reader that is familiar with the control-as-inference framework might notice the bias issue as soon as reading the algorithm box for Algorithm 1 and will be confused about why this problem is not addressed. To be clear, it is fine to assume deterministic systems, but if an algorithm has an obvious pitfall, it should be discussed more prominently.
* I'm not sure that the systems in the paper are actually deterministic. The toy experiment features additional agents which "[...] move adversarially towards the ego agent with some added Gaussian noise." (Section 4.1). This sounds like a stochastic system to me, unless the movement of the adversaries is pre-computed and then kept fixed throughout.

---

[2] Reinforcement Learning and Control as Probabilistic Inference: Tutorial and Review, https://arxiv.org/abs/1805.00909

**Strength And Weaknesses:**

**strengths**

* The paper is proposing a simple modification of regular SMC (in the context of planning) which brings two benefits: a computational one (that we can use a larger number of particles for the same number of transition evaluations) and a conceptual one (that the resulting algorithm can be used in a model-free fashion).
* The paper contains results on refining a prior policy, which is an interesting scenario that is somewhat underrepresented in current research.

**weaknesses**

* Given the method's proximity to [1], it is unfortunate that the paper doesn't contain a comparison to [1] in the same experimental suite or an overlapping one. That makes it harder to compare the two methods.
* The baselines considered in the experiments are: ITRA, SAC, SMC and rejection sampling. The only baseline which has previous published results on these problems is ITRA, which the proposed method does outperform, but that is also to be expected since the authors are building on top of ITRA. In other words, the paper is missing a baseline which has previous published results on this experiment suite. In my opinion this is the paper's biggest issue at the moment.

---

[1] Probabilistic Planning with Sequential Monte Carlo Methods, https://openreview.net/forum?id=ByetGn0cYX

**Summary Of The Paper:**

The paper proposes a sequential monte carlo (SMC) algorithm for planning. The main contribution is to define the particle weights using the Q(s, a)-function, which allows using a large number of actions for the same state without having to evaluate the transition model for each.

**Summary Of The Review:**

The paper proposes a simple method which has tangible benefits. The presentation could be improved to make limitations clearer (i.e. the optimism bias) and the experimental evaluation could be strengthened (e.g. by comparing to [1] using parts of their experiments section) but the paper still makes contributions that are worthy of acceptance.

---

[1] Probabilistic Planning with Sequential Monte Carlo Methods, https://openreview.net/forum?id=ByetGn0cYX

---

> ### Author Response · Authors · 2022-11-16
> **Response to Reviewer 9uTg**
>
> We would like to thank the reviewer for their feedback. Please find our response to each of your points below.
>
> - **Regarding the proposed weaknesses**
>
> We understand the reviewer’s concerns regarding our omission of Mujoco in the set of experiments included in our paper, especially because it was the primary experiment in Piché et al. (2019). We do not include these experiments for a number of different reasons, first Piché et al. (2019) does not include code alongside their submission, which makes the exact reproduction of their results very difficult. Second, as was mentioned in the response to Reviewer qbd3, a leading motivation for CriticSMC is to take advantage of existing powerful implicit probabilistic models. Thus in our experiments, we attempt to extract behavior that satisfies certain conditions while staying as close as possible to the prior policy. In the autonomous vehicle setting, this can easily be tested by learning to avoid committing collisions with other agents while still behaving realistically, relative to the selected prior behavioral model ITRA. Testing this type of behavior in a locomotion setting would be significantly harder, as it would require access to a prior over human movement, which is not easily available in the standard benchmark.
>
> - **The way the optimism bias is discussed is not ideal, because a reader that is familiar with the control-as-inference framework might notice the bias issue as soon as reading the algorithm box for Algorithm 1 and will be confused about why this problem is not addressed. To be clear, it is fine to assume deterministic systems, but if an algorithm has an obvious pitfall, it should be discussed more prominently.**
>
> We thank the reviewer for highlighting this point of confusion. We have corrected Sections 2 and 3 (as well as the corresponding mathematical notation used throughout the paper) to clarify that we assume deterministic state transitions throughout. We also responded to a similar question from Reviewer qbn3 which answer we think is useful to further clarify any concerns regarding stochastic/deterministic state transitions. We repeat below our response for the reviewer’s convenience.
>
> We would like to emphasize that an extension of CriticSMC to support stochastic state transitions is relatively straightforward to derive, though learning a soft-Q function with stochastic state transitions can be challenging in practice due to optimism bias (see Section 6).  In the case where the expectation of the next state given the current state and action can be computed efficiently, or the density of the state transition distribution is known, one can obtain unbiased soft-Q estimates. In order to take advantage of both SMC and the soft-Q estimates, we would also require an explicit density function over the state transitions to correctly reweight particles. Because our experiments are restricted to deterministic environments, we have modified our notation to explicitly use deterministic transitions, and leave an in-depth analysis of our methodology in stochastic environments for future work. We emphasize that Piché et al. (2019) also describe a general framework, but like our work, only consider environments with deterministic transition dynamics in their experiments.
>
> - **I'm not sure that the systems in the paper are actually deterministic. The toy experiment features additional agents which "[...] move adversarially towards the ego agent with some added Gaussian noise." (Section 4.1). This sounds like a stochastic system to me, unless the movement of the adversaries is pre-computed and then kept fixed throughout.**
>
> We thank the reviewer for pointing out this inconsistency. We re-ran all the experiments that use the toy environment with deterministic behavior for the adversarial non-ego agents in the environment. We updated the manuscript with the new results and corrected the description of the environment. We would like to point out that although the environment was stochastic, CriticSMC was able to successfully learn a useful heuristic factor expressed as a soft-Q function that significantly improved the conditional inference performance of SMC. The reason is that the source of stochasticity was provided as noise by a normal Gaussian distribution. We expect the learning method to suffer from optimism bias in more realistic stochastic environment settings.

---

### Official Review · Reviewer_qbn3 · 2022-10-25

**Confidence:** 3
**Correctness:** 3
**Technical Novelty And Significance:** 3
**Empirical Novelty And Significance:** 3
**Recommendation:** 6

**Clarity, Quality, Novelty And Reproducibility:**

* The paper is mostly well written and the ideas seem novel.

**Strength And Weaknesses:**

# Strengths
* The idea of using Q-functions to guide sampling in SMC when observations are sparse is interesting and might be useful beyond the RL and planning contexts.
* The proposed algorithm shows significant performance improvements in a toy collision-avoidance experiment and a more realistic one using a benchmarking dataset when compared to a baseline human-like driving policies and soft-actor critic.

# Weaknesses
* **Background.** The background on SMC for planning-as-inference is quite confusing for someone familiar with SMC and control, but unfamiliar with this line of work applying SMC to planning. Questions I had at a first pass: *Is it only simulating states in parallel and "observing" the optimality indicator*? *How do you use the sampler with a (physical) agent*? There is no clear distinction between **what's done in simulation and what's applied in the actual environment** by the agent. I had to go and read Piché et al.'s (2019) work on SMC for planning to understand that SMC is playing the role of a sampler from the optimal state-action trajectories distribution. The sampled optimal action trajectories can then either be executed blindly by an agent in the target environment after sampling or be used in a model predictive control (MPC) fashion where only the first action is applied to the environment, and from the resulting next state a new SMC planning procedure unrolls.

* **Sec. 3.1: Deterministic vs. stochastic state transitions.** The soft Q-function approximation in Equation 3 relies on the assumption of deterministic state transitions, but the algorithm descriptions in both Figure 2 and Algorithm 1 use stochastic state transitions. It also gives the impression to me that deterministic transitions could lead to some degenaracy in the SMC scheme. However, this mixed presentation in the methodology is confusing and does not explain how deterministic transitions may affect sampling performance in practice.

* **Sec. 3.2: Data for training.** It is not clear to me what is the used to learn the parametric Q function estimators in Eq. 4. No reference for what *state-action occupancy* (SAO) means is provided. I am not sure if it is based on the experience data (from the target environment) observed up to a given point in time, when executing online, or purely on simulation data using the world model that is applied within SMC.

* **Related work.** It might be worth contrasting the paper with some earlier work on SMC for RL:

    Lazaric, A., Restelli, M., & Bonarini, A. (2007). Reinforcement Learning in Continuous Action Spaces through Sequential Monte Carlo Methods. In J. Platt, D. Koller, Y. Singer, & S. Roweis (Eds.), Advances in Neural Information Processing Systems (Vol. 20).

* **Experiments.** The method is proposed as a general framework for planning in environments with sparse rewards, but it is only tested on two autonomous driving environments. It would be beneficial to compare its improvements over state-of-the-art on RL methods for sparse rewards in other classic environments used in this line of work, such as Atari games. Otherwise, the claims of the paper should at least be adjusted to reflect the focus on autonomous driving.

**Summary Of The Paper:**

This paper proposes an approach for planning-as-inference in reinforcement learning problems with sparse rewards. A method based on sequential Monte Carlo is derived, similar to Piché et al. (2019), to sample from the optimal state-action trajectories distribution by applying filtering and smoothing techniques. The algorithm uses estimates of a soft Q-function as a critic to provide heuristic factors when resampling actions according to their optimality within the SMC loop. In contrast to prior work on SMC for planning in RL, CriticSMC uses a fixed prior policy, instead of learned one, and uses the learned Q-function to guide action sampling, which simplifies the training process.

**Summary Of The Review:**

The paper's contribution seems novel, but aspects of the methodology are confusing and the experimental comparisons should be expanded to better support the claims.

#### Rebuttal update
I have raised my score (5 to 6) after reading the authors' response.

---

> ### Author Response · Authors · 2022-11-16
> **Response to Reviewer qbn3 (Part 1)**
>
> We would like to thank the reviewer for their feedback. Please find our response to each of your points below. We note that all changes to our paper have been highlighted in green to make changes within the updated draft more apparent.
>
> - **Regarding the “Background” comment, we will break down a couple of the questions posed by the reviewer**
>
>     - _"Is it only simulating states in parallel and "observing" the optimality indicator? How do you use the sampler with a (physical) agent?"_
>
>         In CriticSMC, there are two possible settings. In setting (1), we have access to a simulator that we can step and reset. This setting is equivalent to the one found in Piché et al. (2019) and is the setting we assume when training the Soft-Q function using SMC. It is crucial to note that any known variant of SMC which relies on restarts or resampling must lay within this setting. The second setting (2) is one in which we do not have direct control over the simulator, and represents the sampler used with a “physical agent”. Here, unlike Piché et al. (2019), we re-use the Soft-Q function heuristic factors and a single environment (namely the one in which the physical agent finds itself) to produce actions and then step the environment which we do not directly control. This is achieved in CriticSMC by assuming the use of one particle and multiple putative action particles.
>
>         As is discussed in Sections 3.3 and 4.2, this second setting can in some sense correspond to the MPC setting mentioned by the reviewer, in which an action is sampled using strictly heuristic factors, the resulting state transition is observed, and a new action is sampled in the same fashion. Alternatively, if we have access to environment dynamics, we could also take multiple steps in the environment after each sampled action, and use the Monte Carlo samples to reduce the bias in our estimate of the expected reward ahead. We emphasize that in both settings, CriticSMC can perform extremely well, as is displayed by our experimental results.
>
>     -  _"There is no clear distinction between **what's done in simulation and what's applied in the actual environment** by the agent."_
>
>         We thank the reviewer for highlighting this lack of clarity and have updated the draft accordingly. We note here, that like Piché et al. (2019), in order to train our critic, we rely on an environment that can be reset (setting (1) from the response above). After the critic has been trained to estimate the soft-Q function of the prior, it can be used in a setting where direct control over the environment is not required (setting (2) from above). Using the terminology of the reviewer, the critic is trained “in simulation” while the evaluation is done in an “actual” environment.
>
> - **Regarding “Sec. 3.1: Deterministic vs. stochastic state transitions”**
>
>     We thank the reviewer for highlighting this inconsistency, which we have addressed in the revised draft. Specifically, we have corrected Sections 2 and 3, as well as the corresponding mathematical notation therein to clarify that we assume deterministic state transitions throughout the paper. We would like to emphasize that an extension of CriticSMC to support stochastic state transitions is relatively straightforward to derive, though learning a soft-Q function with stochastic state transitions can be challenging in practice due to optimism bias (see Section 6).
>
>     In the case where the expectation of the next state given the current state and action can be computed efficiently, or the density of the state transition distribution is known, one can obtain unbiased soft-Q estimates. In order to take advantage of both SMC and the soft-Q estimates, we would also require an explicit density function over the state transitions to correctly reweight particles. Because our experiments are restricted to deterministic environments, we have modified our notation to explicitly use deterministic transitions, and leave an in-depth analysis of our methodology in stochastic environments for future work. We emphasize that Piché et al. (2019) also describe a general framework, but like our work, only consider environments with deterministic transition dynamics in their experiments.

---

> ### Author Response · Authors · 2022-11-16
> **Response to Reviewer qbn3 (Part 2)**
>
> - **Regarding “Sec. 3.2: Data for training”**
>
>     The state-action occupancy (SAO) induced by CriticSMC refers to the set of all state-action pairs visited by the policy induced by the CriticSMC algorithm up to and including that point in time. This is to say, if we have interacted with the environment for a total of 100 steps, and updated the critic 100 times, the SAO buffer will include a sequence of 100 state-action tuples (along with their reward) generated using a sequence of the first 100 different Q functions. Crucially, the SAO buffer is not generated from sampling actions strictly under the prior but instead is generated using the distribution induced by the particle filtering algorithm defined in Algorithm 1.
>
>     We also would like to be clear with the reviewer, in our setting, there is no distinction between the environment, and the so-called “world model”. In our setting, as is mentioned in the main text, we assume access to a state transition function (e.g. environment or simulator), in order to execute Algorithm 1. Incorporation of a world model approximating the dynamics of the state transition function into this framework would be trivial, provided the model itself could be trained efficiently and accurately online. We leave this extension for future work and focus on the behavior models explicitly generated by accurate, physically realistic driving simulation.
>
> - **Regarding “Related work”**
>
>     We thank the reviewer for informing us of this related work. Although it is distinct from our own, it adds a more complete view of the role inference and sequential Monte Carlo has to play in reinforcement learning. We have included the related work in Section 5 but describe it briefly here as well. Unlike our work, the authors of Lazaric et al. (2007) do not consider the RL as inference framework in order to produce their resulting SMC-control algorithm. This is important, as their resulting algorithm varies from our own in two important ways. First, they do not rely on the policy induced by the softmax action distribution, but instead directly learn a policy in a similar fashion to Piché et al. (2019). Second, in learning this policy distribution, instead of sampling under the support of the expert, like Piché et al. (2019) do not take advantage of strong prior distributions. We also note, that like Piché et al. (2019) also evaluate their algorithms in a low-dimensional control environment while we evaluate our models in a high-dimensional image-based setting.
>
> - **Regarding “Experiments”**
>
>     We claim that CriticSMC forms an extension to standard SMC algorithms for state-space models by utilizing heuristic factors expressed as future likelihoods, which make the overall computation of these weights significantly more efficient. This efficiency is especially apparent in sparse reward settings, like the ones discussed in our paper. While additional experimental results can always improve understanding of the benefits and limitations of an algorithm, we believe that given the experiments provided by the literature most closely related to our own (Piché et al. (2019), Lazaric et al. (2007)), we have more than proven the utility of our framework in a high impact area of research (autonomous driving).
>
>     Additionally, a leading motivation for CriticSMC, is to take advantage of existing powerful implicit probabilistic models (e.g. ITRA [Scibior et al. (2021)]) to extract behavior that satisfies certain conditions while staying as close as possible to the prior policy. In our experiments, this is tested by learning to avoid committing collisions with other agents while still behaving realistically, relative to our prior behavioral model ITRA. Thus, when considering classical RL benchmarks like Atari or Mujoco, we see that there is not a specific prior distribution that we would clearly want to encode into the target distribution. While we could contrive such a benchmark using examples collected from human players, we believe such an experiment would be largely redundant with our own and falls outside of the scope of this paper.

---

### Official Review · Reviewer_toeN · 2022-10-30

**Confidence:** 3
**Correctness:** 2
**Technical Novelty And Significance:** 2
**Empirical Novelty And Significance:** 1
**Recommendation:** 1

**Clarity, Quality, Novelty And Reproducibility:**

The paper is reasonably well written.
The novelty of the work seems extremely limited. Every modification presented in this approach seems to be already present in the previous literature.
I cannot evaluate the reproducibility of this approach at this stage.

**Strength And Weaknesses:**

- The paper is sufficiently well written
- The paper is too incremental
- The paper lacks a proper experimental analysis on standard and complex benchmarks: the toy example is too simple, and the simulator seems too complicated and convoluted to understand the algorithm's behavior.
- it is missing a strong theoretical contribution to the method, right now it seems to be mostly a heuristics
- The paper builds on pretty strong assumptions: known model and deterministic dynamics
- The paper tries to tackle a typical problem of safe reinforcement learning, without citing any of the approaches and papers on the areas.

**Summary Of The Paper:**

This paper proposes the Critic Sequential Monte-Carlo approach, a simple extension of the Sequential Monte-Carlo method, in the framework of "planning as inference".
The approach combines the sequential Monte-Carlo technique with heuristic factors computed using a soft q-function.
The authors then test the framework in two safety-related tasks, a multi-agent point-mass environment, and a driving simulator.


**Summary Of The Review:**

While this paper seems to be overall a decent contribution, it is too incremental. To accept incremental work, I would expect clear performance improvements, a strong experimental campaign, vast ablation studies, or interesting insights from the theoretical standpoint.
This paper misses all the points above.

Furthermore, the authors are using safety scenarios as a test bed. There is an entire literature on safety in Markov decision processes:
1) The Constrained MDP framework (CMDP)
2) The Control barrier function approach
3) The shielding approach
4) Reachability analysis

Many of these methods are also model-based and are able to ensure safety while optimizing a policy. It is easy to perform slight modifications to these methods and could fit perfectly the target domains (I suspect these methods will outperform the proposed approach in the given tasks)

I suggest the reviewer either expand the set of tasks to show more generality of their approach (they claim it, but is not shown) or add safe RL baselines to the comparison. For sure, if the focus of the paper is safety, at least a discussion on safe RL methods should be fundamental, explaining why a "planning as inference" approach should be better than methods explicitly designed to enforce safety.

---

> ### Author Response · Authors · 2022-11-17
> **Response to Reviewer toeN (Part 1)**
>
> We would like to thank the reviewer for taking the time to consider our work. We will first address some of the comments made in the strengths and weaknesses section.
>
> - **The paper lacks a proper experimental analysis on standard and complex benchmarks: the toy example is too simple, and the simulator seems too complicated and convoluted to understand the algorithm's behavior.**
>
> We are not sure how to address this, as it asserts that our experiments are both too simple and too complicated at the same time. If the reviewer means to say that the baselines which were considered in relation to existing work [Piché et al. (2019)] make comparison difficult, we can address this concern, which was also posed by Reviewer qbn3 and Reviewer 9uTg. We invite the reviewer to read the corresponding responses. For convenience, we reiterate the main arguments. CriticSMC was produced to solve conditional inference problems in settings that satisfy the following characteristics. First, we assume access to a strong, or behaviorally relevant prior distribution (i.e. for autonomous vehicles this is a prior over human-like behavior). Second, we assume that sampling from the state space is expensive (i.e. in the autonomous driving setting, this requires expensive, physically realistic state-transition dynamics, and high-resolution image rendering). Lastly, the problem we are interested in induces a reward that is sparse while trying to stay as close as possible to the prior (i.e. committing no collisions while driving in a human-like way). The benchmark experiments provided by Piché et al. (2019) do not satisfy these characteristics. Thus we believe they are not well suited to illustrate the utility of CriticSMC, which was applied in a more challenging (high-dimensional and sparse) environment, that is closer to realistic industrial applications.
>
> - **It is missing a strong theoretical contribution to the method, right now it seems to be mostly a heuristics.**
>
> We make no claims about strong theoretical contributions to the area of optimal control, however, CriticSMC is an inference algorithm for state-space models. It builds upon sequential Monte Carlo (SMC) to make conditional inference more computationally efficient. This means that we inherit the asymptotic convergence guarantees of SMC targeting the correct posterior (under standard assumptions). Crucially, it is reductive to describe our work and the CriticSMC algorithm as randomly chosen heuristics. Our work represents a non-trivial modification to one of the most popular general inference algorithms available (SMC) and makes it more sample efficient in settings where the soft-Q function can be accurately estimated.
>
> - **The paper builds on pretty strong assumptions: known model and deterministic dynamics**
>
> The CriticSMC algorithm does not require a model of the world. It takes as input a state transition function which can be represented by the environment directly or a world model/simulator. If the state transition function permits stepping and resetting multiple times during an episode, CriticSMC can take advantage of that and use multiple particles along with multiple putative action particles (see Section 3.3). If that’s not the case such as when we act directly in the environment, CriticSMC can be used in a model-free online control fashion (see Section 4.2) by taking advantage of a large population of putative action particles.
>
> Regarding deterministic dynamics, we responded to a similar question from Reviewer qbn3 which we repeat below for the reviewer’s convenience.
>
> An extension of CriticSMC to support stochastic state transitions is relatively straightforward to derive, though learning a soft-Q function with stochastic state transitions can be challenging in practice due to optimism bias (see Section 6). In the case where the expectation of the next state given the current state and action can be computed efficiently, or the density of the state transition distribution is known, one can obtain unbiased soft-Q estimates. In order to take advantage of both SMC and the soft-Q estimates, we would also require an explicit density function over the state transitions to correctly reweight particles. Because our experiments are restricted to deterministic environments, we have modified our notation to explicitly use deterministic transitions, and leave an in-depth analysis of our methodology in stochastic environments for future work. We emphasize that Piché et al. (2019) also describe a general framework, but like our work, only consider environments with deterministic transition dynamics in their experiments.

---

> ### Author Response · Authors · 2022-11-17
> **Response to Reviewer toeN (Part 2)**
>
> Next, the reviewer suggests we compare CriticSMC with methods from the safety in Markov decision processes literature. Because the reviewer does not provide any references to any paper in the suggested four broad categories of algorithms in safe RL, we consider four examples of these algorithms which we found below. The reviewer claims that each of these algorithm classes functions under the same conditions and assumptions as CriticSMC. However, as we explain below, each method differs from CriticSMC in one or more important ways.
>
> - **The Constrained MDP framework (CMDP)**
>
> [Safe Reinforcement Learning in Constrained Markov Decision Processes, Wachi et al. (2020)](https://arxiv.org/abs/2008.06626)
>
> This method is not an inference method (unlike CriticSMC) and does not take advantage of existing powerful prior policy models such as realistic human-like driving behavior models. In this setting, the SNO-MDP algorithm operates in an alternating fashion. First, it learns the safety constraints by exploring directly under known low-dimensional dynamics. Once the safe region has been found, the algorithm then maximizes a reward subject to staying within the constrained (safe) subset of the environment. Designing a reward scheme for learning RL agents which additionally exhibit realistic human-like driving behavior while staying within the constraint set, is outside the scope of this work [Wachi et al. (2020)] . To drive home the differences between our work and this one, we consider explicitly the assumptions and settings assumed by our work and Wachi et al (2020):
>
> The list of assumptions made by the CMDP paper includes:
>
>     - Deterministic state transitions
>     - Non-sparse reward scheme
>     - Discrete action space
>     - Low-dimensional grid world state space
>
> while we show in our experiments that CriticSMC can work with:
>
>     - Deterministic state transitions (but can be extended to stochastic environments if optimism bias is appropriately mitigated)
>     - Sparse rewards that can only occur at the end of an event or episode
>     - Continuous action spaces (see Section 4 in the paper)
>     - High-dimensional partially observable states (Section 4.2 which uses two 256x256x3 birdview images as states)
>
> - **The shielding approach**
>
> [Safe Multi-Agent Reinforcement Learning via Shielding, Elsayed-Aly et al. (2021)](https://arxiv.org/abs/2101.11196)
>
> The method proposes to learn “shields” that will directly modify the deterministically generated actions made by an agent. Here, the shield will pick an action based on some safety specifications, then a penalty value will be returned to the agent and the action of the shield will be executed in the environment. The shield is expressed as a deterministic finite automaton (DFA) which is created based on some assumed coarse environment abstraction. This approach works for low-dimensional discrete state and action environments but is completely intractable in the driving simulation environment we consider. Additionally, it cannot perform conditional inference from powerful existing probabilistic prior policies. The purpose of CriticSMC is not to arbitrarily modify an action picked by the prior model to make it safe. Instead, CriticSMC aims to sample multiple possible actions under the prior model and evaluate the possibility these actions will commit an infraction in the future.
>
> - **Reachability analysis**
>
> [Safe Reinforcement Learning Using Black-Box Reachability Analysis, Selim et al. (2022)](https://arxiv.org/abs/2204.07417)
>
> Similarly to the shielding approach, the idea here is to directly adjust “unsafe” actions made by the RL policy. The adjustment is performed based on reachability sets of regions in the environment that are reachable according to a known safety function. All experiments are completed in low-dimensional control settings which are not comparable to our own. Additionally, knowledge of this safety function is not assumed in our setting and is unrealistic in the autonomous vehicle setting, where the internal dynamics of the models can be significantly more complicated. Lastly, this method does not seem to take advantage of any powerful informative prior distribution over actions to produce safe and realistic behavioral models, but instead simply forces online interaction of a learned RL agent to be safe through reachability analysis which is itself domain specific.

---

> ### Author Response · Authors · 2022-11-17
> **Response to Reviewer toeN (Part 3)**
>
> - **The Control barrier function approach**
>
> [Safe Reinforcement Learning Using Robust Control Barrier Functions, Emam et al. (2021)](https://arxiv.org/abs/2110.05415)
>
> Similarly to the shielding and reachability analysis approaches, the control barrier function approach effectively adds an additional safety layer that alters any potentially unsafe actions made by agents to satisfy safety conditions posed by the shielding function. Like the previous approach, the detection of whether or not an action is safe is done in a problem-specific fashion, using an explicit understanding of the model dynamics. For some clarity, CriticSMC does not learn a world model and does not assume that the environment or the simulator follows any specific structure (aside from the determinism already discussed). This boils down to only assuming that the simulator can produce a next state given a state and action pair, instead of assuming that the internal integration of the simulator follows specific characteristics. For this reason, as well as the dimensionality and prior-behaviors discussion mentioned in the previous three methods, this paper cannot be compared to our own work.\
> \
> \
> We conclude by directly addressing some of the more general review comments below. Starting with the following comment:
>
> - **“Many of these methods are also model-based and are able to ensure safety while optimizing a policy. It is easy to perform slight modifications to these methods and could fit perfectly the target domains (I suspect these methods will outperform the proposed approach in the given tasks)”**
>
> As was discussed in the brief literature review given above, every method which we investigated in the four subfields you listed varied from our own work in a number of different ways. First, these methods rely on not only the ability to sample or compute the next state from the environment and then reset, but also very crucially, make assumptions about the underlying dynamical system which is producing the next state. The assumptions made on this dynamical system are what allow these authors to produce the functions which take actions sampled from the policy, and return “safe” versions of these actions. We do not assume access to such functions, and thus it is not clear to us how slight modifications would allow these methods to perform equally well as CriticSMC under the same assumptions. Second, the goal of our work is to create infraction-free realistic driving simulations. None of these authors produce algorithms that are targeted toward behavioral realism. Third, all algorithms considered operate on low-dimensional spaces not comparable to our own, high-dimensional image-based setting.
>
> - **“While this paper seems to be overall a decent contribution, it is too incremental. To accept incremental work, I would expect clear performance improvements, a strong experimental campaign, vast ablation studies, or interesting insights from the theoretical standpoint. This paper misses all the points above.”**
>
> We want to thank all reviewers for their thoughtful and considerate comments and suggestions. However, we want to push back against comments like this. Although we recognize the intent behind them, the lack of specificity makes it very difficult to apply improvements to our own work, or respond for clarification. If the reviewer can provide a specific work that is proven to tackle the problem that CriticSMC solves under the same motivation, assumptions, and experimental settings, we will be more than happy to include a discussion of this work. Unfortunately, the claim that slight modifications of existing safe-RL methods can outperform our own, is too vague for us to correctly respond to (though we have done our best in the section above). Similarly, we have included a number of ablation studies in our paper (see Section 4.3), however, including a “vast” number of additional, unspecified ablation studies is obviously impossible. If the ablation studies already included in the main paper, which test the design choices of CriticSMC are not sufficient, please notify us of what additional studies should be included, and why they are relevant to our algorithm. We have done our best to interpret the issues raised by the reviewer, but without specific examples that support the claims made, it is difficult for us to understand how to take these comments and improve our work.

---

### Official Review · Reviewer_ViDn · 2022-10-30

**Confidence:** 3
**Correctness:** 3
**Technical Novelty And Significance:** 3
**Empirical Novelty And Significance:** 3
**Recommendation:** 6

**Clarity, Quality, Novelty And Reproducibility:**

As stated, the main point of the paper comes out very clearly but some of the auxiliary points are not as clear. For example, I didn't follow the extension to human-like driving behavior. The paper discusses using a baseline model ITRA, but it doesn't go into enough detail of how this model was merged/improved with the Critic SMC algorithm in the current paper.

The main idea of critic SMC, on the other hand, appears easy to reproduce.

As far as novelty this paper is a logical extension of prior work, as explained earlier. However it is still distinct from previous work.

The quality of the writing and experiments is quite good. In particular the state, action visualization in Figure 3 very clearly demonstrates the main contribution of this work.


**Strength And Weaknesses:**

The paper is written quite clearly and the main point is very easy to grasp. This is a very logical extension of the previous work by Piché et al 2019 which learns the soft-state-value function. The idea of using putative particles to learn the soft-state-action-value function aka the critic is quite well motivated.

What I have would have liked to see is some tradeoffs between using a large number of putative particles versus using a few putative states. In the work of Piché only one putative state is used, but one could consider a simple extension of that approach where multiple putative states are used. I understand the paper's contention that putative actions are cheaper than putative states, but putative actions are not entirely free either.

Nevertheless, there are clear benefits of doing model-free learning unlike the previous approach of Piché.


**Summary Of The Paper:**

The paper proposes a solution to the planning as inference problem in domains where it is easy to generate samples of actions in the current state but it is expensive to simulate samples of next states. The proposed solution involves learning a parametric approximation of the soft value of each state-action pair by using gradient descent. The paper also shows some examples on a self-driving car datasets and demonstrate better results. There are also some ablation studies.

The main contribution of the work is the observation that in some domains it is hard to simulate the next state, but very cheap to sample an action in the current state. This implies that the existing methods in the literature of learning a soft value of a state is not as helpful as learning the soft-value of a state-action pair (the so called critic function). The authors demonstrate this point by devising a particle filter-based algorithm that uses hypothetical particles in the training episodes to learn the critic function quickly.

**Summary Of The Review:**

Slightly novel contribution in the area of planning as inference using deep learning with some parts unclear, but in general a reasonable contribution.

---

> ### Author Response · Authors · 2022-11-15
> **Response to Reviewer ViDn**
>
> We would like to thank the reviewer for their feedback. Please find our response to each of your points below.
>
> - **What I have would have liked to see is some tradeoffs between using a large number of putative particles versus using a few putative states. In the work of Piché only one putative state is used, but one could consider a simple extension of that approach where multiple putative states are used. I understand the paper's contention that putative actions are cheaper than putative states, but putative actions are not entirely free either.**
>
> It’s not completely clear in the context of the paper what “putative states” correspond to, however, we interpret the reviewer's question in the following way. According to our understanding, putative states would correspond to possible next states sampled from the environment given the current state and action.  In our work, we assume deterministic transition dynamics.  This means that we cannot use “putative states” according to the definition given in Section 3.3 because these next state samples would always be the same given the same current state and action pair.
>
> We emphasize that the work of Piché et al. (2019) also evaluates their method in environments with deterministic transition dynamics.  Additionally, the environments evaluated by Piché et al. (2019), have two important distinctions from our own. First, they are evaluated using state spaces defined by low-dimensional vectors, while we evaluate our framework in a realistic driving simulator whose state space is defined by high-dimensional, stacked, RGB images.  Second, our benchmark uses sparse reward feedback, which given the enlarged state space represents a significantly more difficult problem class. With context in mind, as well as the results from Table 3 which display the importance of using a large number of particles, we argue that the way in which we have defined putative particles is an important aspect of the success of CriticSMC.
>
> - **As stated, the main point of the paper comes out very clearly but some of the auxiliary points are not as clear. For example, I didn't follow the extension to human-like driving behavior. The paper discusses using a baseline model ITRA, but it doesn't go into enough detail of how this model was merged/improved with the Critic SMC algorithm in the current paper.**
>
> ITRA [Scibior et al. (2021)] is a probabilistic model trained from human traffic demonstrations to imitate human driving behavior. It takes as input an overhead birdview image centered and rotated according to the agent of interest and proposes acceleration and steering actions. These values are given to an environment that is based on a bicycle kinematic model, which will advance the environment by one timestep, and produce the next birdview state image. We followed the same process for pre-training ITRA as Scibior et al. (2021). In our reinforcement learning setting, we control one agent called the “ego-agent”, while all other agents in the environment are replayed according to prerecorded demonstrations from the INTERACTION dataset.
>
> We emphasize that CriticSMC is completely agnostic to the generative model used. This is especially important for not only using the ITRA prior but also many other deep generative models whose score function (log-probability) cannot be directly evaluated (GANs, VAEs, etc.).  Additionally, unlike many standard reinforcement learning frameworks, CriticSMC can actually take advantage of strong prior knowledge of the problem. ITRA, which represents a reasonable baseline on its own, is, therefore, a good choice of a prior to display utility of CriticSMC.

---

### Decision · Program_Chairs · 2023-01-20

**Decision:**

Accept: poster

**Justification For Why Not Higher Score:**

The paper provides a contribution with the proposed method, but all reviewers agree the paper could provide better evidence for the method and discussion.

**Justification For Why Not Lower Score:**

The reviewers were split, three for accept and one strongly against.  The detailed rebuttal by the authors and the final agreement of most reviewers indicate that the core contribution of the paper is sufficiently convincing for most readers.  The objection of one reviewer is noted.  The objection stems mostly from the choice of demonstration domain as a safe RL autonomous driving situation, which was complex and not the best experiment to showcase this method as there is a rich literature of specific methods for such domains.

For researchers engaged with SMC methods in general simulation domains, this paper provides a useful contribution. The area chair is in favor of accepting the paper.  The AC recommends that the authors use suggestions from the reviews to improve the final paper.

**Metareview: Summary, Strengths And Weaknesses:**

Summary:

This paper presents CriticSMC, a sequential Monte Carlo (SMC) method for reinforcement learning under the planning as inference view.  Concretely, this work extends the work of Piche et al (2019), by using a soft Q function to construct a proposal distribution for re-weighting particles associated to an afterstate (after the action has been selected but before the next state is observed).  Using the soft Q function to modify particle-selection enables a more efficient use of available compute in several domains (where evaluating an action value can be substantially cheaper than propagation through a transition dynamics function).  The method was demonstrated on a toy domain and a driving simulator.

Strengths:

The overall idea is simple.  After resolving questions about the problem setup and goals, three reviewers were in agreement that this work provides a useful method.

Weaknesses:

The reviewers also noted several weaknesses in the paper.  One reviewer was concerned about the limited treatment of safety constrained RL, which was the domain used to showcase the method, but where other techniques from that literature were ignored.  Multiple reviewers were concerned by the fact that the overall approach suffers from an overoptimism bias coming from the use of deterministic models, which is well known and was not given adequate treatment in the original paper.  A final concern lies in the limited experimental reproducibility and empirical evidence from the one complex driving simulator and the one toy domain.

**Note From Pc:**

if the above contains the word "oral" or "spotlight" please see: "oral" presentation means -> notable-top-5% and "spotlight" means -> notable-top-25%. As stated in our emails, we are disassociating presentation type from AC recommendations